

# Mediterranean cyclones: Current knowledge and open questions on dynamics, prediction, climatology and impacts

Emmanouil Flaounas[1], Silvio Davolio[2], Shira Raveh-Rubin[3], Florian Pantillon[4], Mario Marcello Miglietta[5], Miguel Angel Gaertner[6], Maria Hatzaki[7], Victor Homar[8], Samira Khodayar[9], Gerasimos Korres[1], Vassiliki Kotroni[10], Jonilda Kushta[11], Marco Reale[12, 13], Didier Ricard[14]

[1]Institute of Oceanography, Hellenic Center for Marine Research, Athens, Greece
[2]National Research Council of Italy, Institute of Atmospheric Sciences and Climate (CNR-ISAC), Bologna, Italy
[3]Department of Earth and Planetary Sciences, Weizmann Institute of Science, Rehovot, Israel
[4]Laboratoire d'Aérologie, Université de Toulouse, CNRS, UPS, IRD, Toulouse, France
[5]National Research Council of Italy, Institute of Atmospheric Sciences and Climate (CNR-ISAC), Padua, Italy
[6]University of Castilla-La Mancha, Faculty of Environmental Sciences and Biochemistry, Toledo, Spain
[7]Department of Geology and Geoenvironment, National and Kapodistrian University of Athens, 10679 Athens, Greece
[8]Meteorology Group, Physics Department, Universitat de les Illes Balears, Palma, Mallorca, Spain
[9]Mediterranean Centre for Environmental Studies (CEAM), Valencia, Spain
[10]Institute of Environmental Research and Sustainable Development, National Observatory of Athens (NOA), Athens, Greece
[11]Climate and Atmosphere Research Center (CARE-C), The Cyprus Institute, Nicosia 2121, Cyprus
[12]National Institute of Oceanography and Applied Geophysics - OGS, Via Beirut, 2, 34151, Trieste, Italy
[13]Abdus Salam International Centre for Theoretical Physics, ICTP, ESP group, Trieste, Italy
[14]CNRM, Université de Toulouse, Météo-France, CNRS, Toulouse, France

*Correspondence to*: Emmanouil Flaounas (em.flaounas@hcmr.gr)

**Abstract.** A large number of intense cyclones occur every year in the Mediterranean basin, one of the climate change hotspots. Producing a broad range of severe socio-economic and environmental impacts in such a densely populated region, Mediterranean cyclones call for coordinated and interdisciplinary research efforts. This article aims at supporting these efforts by reviewing the status of knowledge in the broad field of Mediterranean cyclones. First, we focus on the dynamics and atmospheric processes that govern the genesis and development of Mediterranean cyclones. Then, we review the state of the art in forecasting cyclones and relevant high-impact weather. Particular attention is given to Mediterranean cyclone tracks and their physical characteristics in current and future climate. Finally, we focus on the impacts produced by cyclones and we outline the future directions of research that would advance the broader field of Mediterranean cyclones as a whole.

## 1 Introduction

The Mediterranean basin is a relatively small region, but of unique and complex geography. It is characterised by a nearly enclosed basin with sharp land-sea transitions, surrounded by high mountain chains. In this unique environment, cyclogenesis is exceptionally frequent, long ago known to be one of the highest in the globe (Petterssen, 1956). The complex



topographical features, and its transitional location between the tropics and the mid-latitudes downstream of the North Atlantic storm tracks, influence Mediterranean cyclones through a large variety of atmospheric processes. Mediterranean cyclones typically present weaker intensities, smaller sizes and shorter lifetimes than mid-latitude cyclones that develop over open oceans (Trigo, 2006; Campa and Wernli, 2012; Flaounas et al., 2014). Nevertheless, Mediterranean cyclones often produce high impact, being responsible for the grand majority of precipitation and wind climate extremes that affect both the

regional environment and the population of more than 480 million.

Over the last decades, the main research efforts on Mediterranean cyclones focused not only on atmospheric and climate sciences, but also on environmental and socio-economic impacts. At international level, the Mediterranean Experiment (MEDEX, 2000-2010; Jansa et al., 2014) strongly contributed to improving our understanding of Mediterranean cyclone

dynamics, modelling and impacts. The lessons learned from MEDEX were of great importance in the planning phase of the field campaigns of the Hydrological Cycle in Mediterranean Experiment (HyMeX, 2010-2020; Drobinski et al., 2014). Indeed, the HyMeX Special Observing Periods are, to date, the most ambitious and comprehensive observing campaigns internationally coordinated to investigate the water cycle in the Mediterranean (Ducrocq et al., 2014). In later years, new modelling capabilities and observational datasets were made available to the scientific community, while new results from

cross-disciplinary studies emerged. Such results also addressed fundamental questions on cyclone interactions with the Mediterranean Sea and aerosols, showing a strong effect of cyclones on the regional environment and its water cycle, in addition to their socio-economic impact.

Scientific research in the broader field of Mediterranean cyclones is currently advancing rather asymmetrically. For instance,

numerous studies have recently been devoted to tropical-like cyclones, known in the scientific literature as medicanes, a portmanteau of the words "Mediterranean hurricanes" (Emanuel, 2005; Miglietta, 2019). However, medicanes represent a very limited number of intense cyclones, which may not correspond to those with the greatest environmental and socio-economic impacts. In addition, the number of studies that focus on the western Mediterranean is higher compared to those focusing on the eastern Mediterranean or on African coasts. Aiming to the advancement of the broader field of

Mediterranean cyclones as a whole, this article provides a review of the knowledge accumulated in the last decades on Mediterranean cyclones in three main sections: Section 2 focuses on cyclone dynamics, particularly on the physical processes that turn cyclones into severe storms. Gaining insights into these processes, sections 3 and 4 review current challenges in modelling and predicting cyclones on weather and climate scales, respectively. Next, section 5 provides an overview of the variety of high-impact weather conditions produced by cyclones, along with the physical processes involved,

and of the climatological trends. Finally section 6 identifies open questions, proposes new research priorities on the topics that need further investigation, and highlights the observational and modelling approaches that will contribute in this direction.





## 2 Mediterranean cyclone dynamics

Mediterranean cyclones are mostly developed by baroclinic instability and affected by diabatic processes similarly to other extratropical cyclones in the mid-latitude storm track. However, Mediterranean cyclones are further influenced by the abrupt land-sea transitions and by the prominent mountain chains that surround the basin. In addition, due to its location between the tropics and the mid-latitudes, the Mediterranean basin has the potential of hosting subtropical or even tropical-like storms. In this section, we review recent advancements in understanding Mediterranean cyclone dynamics, the wide variety

of involved atmospheric processes and their complex interactions with topography.

### 2.1 Large scale forcing to Mediterranean cyclogenesis

The formation of intense Mediterranean cyclones is typically related to southward deviations of the Polar Jet that causes air masses with high potential vorticity (PV) to intrude into the Mediterranean region. This triggers baroclinic instability similarly to extratropical cyclones over open oceans (Flocas, 2000; Trigo et al., 2002; Nicolaides et al., 2006; Fita et al.,

2007; Claud et al., 2010; Kouroutzoglou et al., 2011; Flaounas et al., 2015). Raveh-Rubin and Flaounas (2017) showed that Rossby wave breaking (RWB) is a common precursor of Mediterranean cyclogenesis. Figure 1 shows composite fields of PV and sea level pressure (SLP) at the time of the mature stage of 200 intense Mediterranean cyclones. A characteristic of RWB in Fig. 1a is an amplified ridge in the midlatitude jet upstream of the Mediterranean (depicted in Fig. 1b by negative PV anomalies overlaid by positive SLP anomalies) and the intruding stratospheric air masses downstream of the Polar Jet

(positive PV anomalies in Fig. 1b). The consequent cyclogenesis is depicted in Fig. 1b by negative SLP anomalies in the central Mediterranean basin.

Stratospheric air mass intrusions southward into the Mediterranean upper troposphere typically manifest as PV streamers (Prezerakos et al., 1996; Homar et al., 2007; Claud et al., 2010; Flaounas et al., 2015). However, it is not uncommon for PV

streamers to disconnect from the stratospheric high-PV reservoir eventually forming a PV cutoff (Homar et al. 2003; Porcù et al., 2007; Fita and Flaounas, 2018; Portmann et al., 2020a). In fact, both PV streamers and cutoffs are common precursors for the development of deep cyclones, including medicanes (Tous et al., 2013; Miglietta et al., 2017). Portmann et al. (2020b) showed that cutoffs in the western Mediterranean typically form in winter under anticyclonic RWB over Europe, and subsequent cyclonic RWB on the cyclonic shear side of the subtropical jet (baroclinic life cycles as described in

Thorcroft et al., 1993). However, several case studies illustrate the possible association of both RWB types with Mediterranean cyclogenesis (e.g., Flocas, 2000; Argence et al., 2008; Raveh-Rubin and Wernli, 2016; Raveh-Rubin and Flaounas, 2017; Portmann et al., 2020a; Givon et al., 2021). When intruding into the Mediterranean, the Polar Jet can merge with the subtropical jet (Prezerakos et al., 2006), similar to the dominant Polar Jet events described in Winters et al. (2020), in which the polar jet merges with the subtropical jet near the climatological location of the latter. This superposition is also

expected to further accelerate both jets (Martius and Wernli, 2012; Raveh-Rubin and Wernli, 2015). Consistently, Flaounas et al. (2015) suggested that the cyclogenesis environment is characterized by strong horizontal shear that, in turn, forces





Mediterranean cyclones to follow a common baroclinic cyclone life-cycle. Compared to cyclones over open oceans, Mediterranean cyclogenesis takes place across lower-tropospheric temperature gradients where baroclinicity is partly due to land-sea surface temperature contrast (Lionello et al., 2006; Trigo et al., 2002). Such baroclinic forcing is however not
expected to significantly enhance the surface cyclone (Flocas, 2000).

## 2.2 Airstreams and fronts

The organization of air masses within the cyclone is key for determining the cyclone lifecycle and its variable impact. The conceptual conveyor belt model integrated by Carlson (1980) for extratropical cyclones includes three cyclone-relative airstreams, namely the warm conveyor belt (WCB), a moist poleward ascending airstream, the cold conveyor belt (CCB),
flowing in the lower troposphere from the cold side of the warm front cyclonically towards the cyclone centre, and the dry air intrusion (DI), which descends slantwise into the cold sector behind the cold front. Based on five case studies, Raveh-Rubin and Wernli (2016) provided a schematic that presents the main features of Mediterranean cyclone dynamics including airstreams, frontal structures, wind gusts and precipitation. This schematic is shown in Fig. 2.

In a first methodological approach to place Mediterranean cyclones in the context of the general conveyor belt model, Ziv et al. (2010) indeed identified the major airstreams in eight representative mature cyclones, although with limited spatial extent compared to oceanic systems. Compositing the 200 most intense cyclones in a 20-year period, Flaounas et al. (2015) found an overall frontal structure and the existence of strongly ascending WCB trajectories (depicted as feature 5 in Fig. 2) and stratospheric DIs (feature 3 in Fig. 2) in almost all cyclones, respectively ahead and behind the PV-streamer (feature 1 in Fig.
2). WCBs have been specifically identified in a number of case studies in the Mediterranean, showing a large case-to-case variability with regard to their spatial extent and contribution to precipitation mainly to the east and north of the cyclone (Flaounas et al. 2016, Raveh-Rubin and Wernli, 2016). Climatologically, Pfahl et al. (2014) found that WCBs are associated with 50-60% of extreme precipitation events in the Mediterranean (blue area in Fig. 2), while Flaounas et al. (2018) highlighted the key role of WCBs, along with deep convection (features 6-8 in Fig. 2), for rainfall intensities. In terms of
strong winds, it is only recently that Brancus et al. (2019) showed the possible presence of sting jets (Clark et al. 2005) in Mediterranean cyclones. Raveh-Rubin and Wernli (2016) identified DI trajectories around five Mediterranean cyclones, encircling the region of the highest winds (feature 4 in Fig. 2). In one case, DIs reaching the cold front were co-located with convective wind gusts and precipitation. On average, cold fronts trailing from Mediterranean cyclones are sharper and longer when they are immediately followed by a dry intrusion, similar to cold fronts trailing from extratropical cyclones over the
main ocean basins (Catto and Raveh-Rubin 2019). However, differently from oceanic fronts, when Mediterranean cold trailing fronts are followed by dry intrusions, frontal precipitation is weaker on average, despite the stronger front intensity and the general correlation between front intensity and precipitation. This is likely a result of the limited moisture supply into the Mediterranean from the south, compared to the North Atlantic or Pacific Oceans.



## 2.3 Role of diabatic processes in cyclones development

After cyclones are formed, baroclinic instability may further intensify surface cyclonic circulation (e.g. Prezerakos et al., 2006; Lagouvardos et al., 2007) and in parallel may favour convection in cyclones centres due to upper-level forcing. Latent heat release is thus expected to deepen cyclones' central SLP and to intensify cyclonic circulation through diabatically-produced PV anomalies in the lower troposphere. The synergistic forcing of baroclinic and diabatic processes to cyclones development is consistent with previous results of Homar et al. (2002) and Fita et al. (2006). The authors applied factor

separation techniques to eleven cases of intense Mediterranean cyclones and showed that the systems' intensification was the result of interactions between upper tropospheric PV streamers and low-level diabatically-produced PV. Such interactions have been also analysed in medicane cases by Cioni et al. (2016) and Fita and Flaounas (2018). The latter showed that, in the December 2005 case, the PV streamer intruding over the Northwest African coast triggered cyclogenesis, but the sharp deepening of the cyclone was related to convection triggered by the PV streamer. A similar scenario was documented in the

well-studied medicane that took place in September 2006: the medicane presented a clear PV-tower structure that was directly related with deep convection (Miglietta et al., 2017), while the explosive intensification was due to the interaction of convection with the upward forcing induced by the PV streamer (Chaboureau et al., 2012).

In a recent analysis, Flaounas et al. (2021) applied piecewise PV inversion to two different components of PV for 100

intense Mediterranean cyclones at the time of their mature stage. These PV components corresponded to the conserved, adiabatically transported PV (dominated by the upper tropospheric PV streamer or cutoff), and to the non-conserved, diabatically-produced PV (dominated in most cases by convection). Figure 3 shows in a scatterplot the contribution of the two PV components to relative vorticity at 850 hPa for each of the 100 cyclones. Results show that diabatic and baroclinic forcings are both typically involved in cyclones mature stage. Furthermore, Fig. 3 shows that the more baroclinically driven

a cyclone is, the less it is expected to be intensified by diabatic processes and vice-versa. In fact, only a small fraction of the 100 cyclones can be considered as being dominantly driven by diabatic forcing (upper end of the scatter plot distribution in Fig. 3). This is consistent with climatological analysis by Galanaki et al. (2016) and Flaounas et al. (2018) who used lightning observations to show that it is only about one third of the most intense Mediterranean cyclones that presented deep convection in their centre during their mature stage. In such cases, convection is expected to significantly contribute to the

development of the cyclones, even to play a leading role to their intensification (Kouroutzoglou et al., 2011, 2012; Raveh-Rubin and Wernli, 2016). In these regards, the availability of humidity within the Mediterranean and from remote sources, the air-sea interactions and the underlying sea surface temperatures (SST) are expected to be crucial for the intensity of convection and its role in cyclones development (Miglietta et al., 2011; Romaniello et al., 2015; Messmer et al., 2017; Pytharoulis, 2018; Flaounas et al., 2019).




## 2.4 Role of the orography in cyclone dynamics

The Mediterranean region presents unique topographical characteristics, with pronounced mountain chains in Northwest Africa and in a long zonally extended area that encompasses the northern side of the basin. Climatological studies clearly showed that Mediterranean cyclogenesis is favoured close to and to the south of mountains, suggesting that the distribution
and frequency of cyclones in the Mediterranean is markedly modulated by the orography (Campins et al., 2011; Buzzi, 2012; Lionello et al., 2016). Although a primary baroclinic disturbance (synoptic scale trough or cyclone) is almost invariably required to provoke cyclogenesis, it is not uncommon that its interaction with the orography favours cyclogenesis in specific preferred locations (Fantini and Davolio, 2001 and references therein) and determines the characteristics of the "secondary" Mediterranean cyclone (Buzzi et al., 2003; Jansa et al., 2014), which often has the characteristics of a lee cyclone (Lin,
2007). A detailed review of Alpine cyclogenesis theories and models has been recently presented by Buzzi et al. (2020).

Although many intense Mediterranean cyclones are not necessarily the fruit of lee-cyclogenesis, this is a distinct feature with respect to mid-latitude cyclones that drew most of the research efforts in the last decades of the last century. A widely accepted conceptual model describes lee cyclogenesis as a two-step process (Buzzi and Tibaldi, 1978). First, as a synoptic-
scale trough approaches the mountain chain, the cold air advection is retarded on the upwind slopes, while on the lee side a positive thermal anomaly and rapid pressure drop is produced, along with wind acceleration (feature 9 in Fig, 2). Second, as the primary wave decouples and crosses the obstacle, baroclinic instability, modified by the presence of the mountain, dominates and leads to a secondary (lee) cyclone that organizes itself, acquiring a larger (synoptic) scale (McTaggart-Cowan, 2010; Tsidulko and Alpert, 2001), while in intense cases diabatic processes as latent heat release, heat surface fluxes
and low-level filaments of PV, generated from the air-flow interaction with the mountains (PV banners; Aebisher and Schar, 1998) can contribute to the formation of low-level PV maxima. Considering this phase by means of the PV-thinking framework, baroclinic cyclone growth occurs as an overlapping and phase locking between the upper level PV streamer and the lower level PV (or thermal) anomaly, as in type-B cyclogenesis (Pettersen and Smebye, 1971).

## 2.5 The special case of medicanes

Medicanes are identified in scientific literature as peculiar cases of Mediterranean cyclones. At first, they were distinguished due to their visual resemblance with tropical cyclones in satellite images. In fact, a Mediterranean cyclone was qualified as a medicane if it presented a central "eye" and spiral cloud bands around the core (Fita et al., 2007; Tous et al., 2013), thus sharing some visual similarities also with polar lows (Rasmussen and Zick, 1987). Such a phenomenological method suggested that medicanes are rare events occurring about 1-3 times per year. Several climatological studies applied various
wind speed criteria and the cyclone phase-space analysis (Hart, 2003) to cyclone tracks from model simulations and reanalysis, confirming the rarity of medicanes (e.g., Cavicchia et al., 2014a; Picornell et al., 2015; Gaertner et al., 2018; Zhang et al., 2020). However, such studies used different methods and datasets, lacking to reach consensus on the physical




definition of medicanes. In contrast to typical tropical cyclogenesis, results from several case studies regard medicanes as baroclinic cyclones that evolve in the mature stage into vortices with structural characteristics similar to tropical cyclones,

i.e. an axisymmetric, deep warm core, generally with a windless center surrounded by strong winds. This is a consequence of the different environmental conditions in which medicanes and tropical cyclones develop. The first difference is the wind shear, which is weak near the tropics while it systematically affects the Mediterranean region due to the southern deviations of the polar jet and the possible presence of the subtropical jet. The second is the SST. Tous et al. (2013) showed that the systems visually identified as medicanes develop over SST of 15-23 °C, well below the threshold of 26.5 °C for tropical

cyclones. In fact, the intrusion of very cold air in the upper troposphere may allow a high conversion efficiency of thermal energy into kinetic energy even in the presence of relatively low SST (Palmén, 1948). This mechanism is also relevant for the subset of tropical cyclones that develop at higher latitudes, over SSTs within the range of 22 to 26.5 °C (McTaggart-Cowan et al., 2015).

## 2.6 Medicanes processes and classification

Due to their rare occurrence, the mechanisms responsible for the development of medicanes were analysed in a relatively limited number of case studies. Typically, the beginning of medicane lifecycles is similar. All systems tend to grow through the interaction of an upper troposphere PV streamer with a low-level baroclinic area, as typically expected for extratropical cyclones. Nevertheless, different evolutions can be identified during maturity when baroclinic forcing, air-sea interactions and convection may all play an important role in cyclones development (Moscatello et al. et al., 2008; Flaounas et al., 2021).

Miglietta and Rotunno (2019) recently categorised medicanes in three main groups with similar dynamical processes during their mature stage (a similar classification was provided recently in Dafis et al., 2020):

Group 1: baroclinic instability plays an important role throughout the cyclones' lifetime and warm seclusion is sufficient to explain the presence of the warm core; nevertheless, most of their intensification can be attributed to convection. Fita and

Flaounas (2018) analysed the medicane of December 2005 and showed that the system presented a clear frontal structure, where the cloudless "eye" was a result of dry air intrusions, similarly to midlatitude storms. Convection was relevant to cyclogenesis and was responsible for the rapid intensification of the system. In contrast, convection was weak during the mature phase of the medicane, when the system was propagating in phase with an upper tropospheric PV cut-off.

Group 2: baroclinicity is relevant only in the initial stage, and, as for tropical cyclones, the theory of wind-induced surface heat exchange (WISHE; Emanuel, 1986; Rotunno and Emanuel, 1987) can explain their intensification through a positive feedback between latent-heat release and air-sea-interactions. However, WISHE may only take place after the occurrence of tropical transition, i.e. after organised convection near the cyclone centre is capable of sustaining the vortex (Davis and Bosart, 2004). For such medicanes, the vertical thermodynamic structure is similar to that of tropical cyclones, as presented

in Fig. 4a for the case of October 1996 (Mazza et al., 2017), and very different from that of a cyclone of group 1 (December




2005 case in Fig. 4b). Medicane Rolf (6-9 November 2011) is another characteristic example: Miglietta et al. (2013) and Dafis et al. (2018; 2020) showed that deep convective clouds occurred mainly during the intensification period, while lightning activity was mostly pronounced about one day before the maximum cyclone strength. This reveals that the spatial and temporal distribution of deep convection and lightning activity is more similar to those of tropical cyclones than to

intense Mediterranean extra-tropical cyclones. Romaniello et al. (2015) showed a significant sensitivity of model simulations to the SST field (weaker cyclone for colder SST).

Group 3: This group includes smaller-scale vortices that develop within the circulation associated with a synoptic-scale cyclone, as in the case of 24-26 September 2006. The latter cyclone is paradigmatic of the different roles that orography can

play in their lifecycle, either favoring cyclogenesis (in the lee of the Atlas mountain in that case), or triggering convection near the system during the interaction with an upper-level PV streamer just before the tropical transition (Moscatello et al., 2008; Chaboureau et al., 2012). A more recent example concerns medicane Qendresa (7-8 November 2014). Bouin and Lebeaupin-Brossier (2020) showed that the deepening and tropical transition of Qendresa, initially developed as an extratropical cyclone, are the result of the synergy between baroclinic and diabatic processes, identifying, as did Pytharoulis

(2018), a strong control of SST over the cyclone intensity. Conversely, Carriò et al. (2017) stressed the importance of the upper-level dynamics to support the tropical transition of the system. In this case, latent-heat release and surface-heat fluxes, as individual processes, seem to be of secondary importance.

Flaounas et al. (2021) highlighted the need to objectively quantify the relative contribution of baroclinic and diabatic

processes to the development of medicanes, showing that both play an important role in the mature stage of several known cases of medicanes. The latter considerations for classification are not unique to hybrid cyclones, such as medicanes, but can be extended to Mediterranean extra-tropical cyclones, considering that sea surface fluxes and latent heat release can significantly modify the cyclone evolution even when baroclinic instability is the main forcing.

## 3 Forecast challenges in Mediterranean cyclones

The close link between Mediterranean cyclones and severe weather phenomena has historically been at the basis of international coordinated efforts devoted to improve the predictability of the broad range of these cyclones. Nevertheless, the medium-range predictability of Mediterranean cyclones is still considered to be relatively limited. This can be attributed to: the multiscale nature of Mediterranean cyclones, the high contrast in the density of observations between terrestrial and maritime areas, and the diversity of geographical features in the basin. Performing more accurate, reliable and timely

forecasts of Mediterranean cyclones is a complex issue that comprises advances in large and convective scale initial condition uncertainty representations, but also in modelling techniques and capabilities. In this section, we review the latest efforts in these two directions.





### 3.1 Dependence of predictability on processes taking place in different scales

As discussed in section 2, an upper-level precursor is essential for Mediterranean cyclogenesis. Such precursors usually
involve RWBs that take place over the central Atlantic Ocean and eventually evolve into PV streamers or cut-offs that
intrude into the Mediterranean region. In this mechanism, diabatic processes in upstream cyclones play an important role in
provoking RWB over the Atlantic (Massacand et al. 2001; Grams et al. 2011; Raveh-Rubin and Flaounas, 2017).
Consequently, the predictability of Mediterranean cyclogenesis strongly relies on the accurate prediction of the wave guide
evolution over the Atlantic Ocean and the processes involved therein.


Several case studies focusing on medicanes have shown that large scale conditions often pose a challenge to the
predictability of Mediterranean cyclogenesis. For instance, Chaboureau et al. (2012) suggested a link between the poor skill
of operational forecasts of the European Centre for Medium-Range Weather Forecasts (ECMWF) for the September 2006
medicane and the extratropical transition of hurricane Helene taking place upstream. The link was confirmed by Pantillon et
al. (2013), who showed that the fraction of ensemble members capturing the medicane formation dropped for lead times
beyond 2 days. This was due to the sensitive phasing between the hurricane and a Rossby wave train, a situation
characterized by high forecast uncertainty as measured by the ensemble spread (Anwender et al. 2008). To gain deeper
insights into the large-scale atmospheric circulation, relevant to the poor forecast of Mediterranean cyclogenesis, Fig. 5
shows the relative locations of the 2006 medicane, hurricane Helene and RWB that eventually triggers cyclogenesis
downstream in the Mediterranean.

A similar pattern was found by Portmann et al. (2020), who showed a large spread of the ECMWF ensemble members in the
position of a Mediterranean PV streamer beyond three days lead time. The spread originated from a short-wave perturbation
over Newfoundland and affected the position and thermal structure of medicane Zorbas (September 2018). These results are
consistent with those of Di Muzio et al. (2019) who used the ensemble forecasts of ECMWF to perform a systematic
analysis of medicanes' predictability. The ensemble members presented a predictability jump with decreasing lead time often
between 5–7 days ahead, suggesting the existence of predictability barriers. The processes and atmospheric conditions that
impose predictability barriers are still an open question. Nevertheless, it seems that Mediterranean cyclogenesis is the final
link of a chain of events of exceptional forecasting difficulty. Indeed, Rodwell et al. (2013) analysed the atmospheric
conditions that led the ECMWF model to poor forecast skill for a series of high impact events over Europe. Composite fields
of the large-scale flow revealed a similar pattern with the one of RWB triggering Mediterranean cyclogenesis, as shown in
Fig. 1b.

These studies raise the need for a better numerical representation of the upper-level waveguide. Detailed observations are
rare and restricted to dedicated field campaigns such as the recent North Atlantic Waveguide and Downstream Impact





Experiment (Schäfler et al. 2018), which documented a case of low predictability associated with a cut-off low triggering heavy precipitation over the Mediterranean. Beyond such rare campaign data, more efforts based on model data are still needed to question the limits of predictability and investigate their case-to-case variability and origin. The representation of cloud diabatic processes is often seen as a source of error growth and forecast uncertainty but initial conditions may still 300 dominate.

## 3.2 Predictability and forecasting strategies

The fundamental role of the upper level dynamics in forecasting western Mediterranean cyclogenetic cases was emphasized by Vich et al. (2011a; 2011b). The authors used two main approaches of ensemble prediction systems (EPS), respectively based on multiphysics simulations and on perturbations to the PV field of the large-scale flow, generated by targeting either 305 zones of adjoint derived sensitivity or of intense values and gradients. All three approaches were evaluated for a series of high-impact cyclones responsible for heavy precipitation. Results showed similar forecast skill among the multiphysics and PV-based approaches, whereas the latter ones were slightly more proficient. In addition, pragmatic methods to generate initial condition perturbations, that target large upper level PV gradients, showed cyclone and precipitation forecast skill equivalent to more complex and computationally expensive adjoint-based methods. Regarding multiphysics approaches, 310 recent studies on medicanes (Miglietta et al. 2015; Pytharoulis et al. 2018; Ragone et al., 2018; Mylonas et al. 2019) showed that tracks, intensity and duration of the cyclones were highly sensitive to convection, microphysics and boundary-layer parameterizations. However, all studies converged to the conclusion that there is no "perfect" model configuration to adequately predict all synoptic-scale characteristics, while the timing of model initial conditions was crucial for the credible representation of cyclones.


The effect of prominent orography on the predictability of cyclone evolution and impacts is a clear example of the complexity of possible nonlinear feedbacks. Flow channeling and the interaction with local topographic features shape the low-level flow and usually determines the triggering, location and intensity of damage-causing convective systems (e.g. Sotillo et al. 2003; Davolio et al. 2016; Amengual et al. 2017; Carrió et al. 2019). Again, the multiscale forcings of 320 geographical features - such as coastal transitions and orographic interactions - pose one of the main current predictability challenges on the evolution of Mediterranean cyclones. Indeed, a wide range of scales is typically relevant to forecast accurately the timing, location, mode and intensity of the damaging impacts (e.g. Delrieu et al. 2005; Lorenzo-Lacruz et al. 2019). Forecast strategies accounting for uncertainties linked to moisture sources and evolution, such as surface fluxes, boundary layer flows and microphysical processes must be investigated to face those challenges.


In a systematic study, the performance of two convection-permitting ensembles (AROME-EPS and COSMO-H2-EPS) has been assessed over the whole HyMeX special observation period SOP1 (Nuissier et al. 2016). Both ensembles exhibited a good probabilistic skill in forecasting heavy precipitation at a relatively high spatial and temporal resolution and were



considered as promising tools for operational forecasts. Another comparison in the HyMeX-SOP1 period among AROME-

EPS, the global PEARP ensemble prediction system, and the AROME-France high-resolution deterministic system, shows
that a high-resolution ensemble (2.5-km horizontal resolution) with few members can outperform a lower-resolution
ensemble (15.5-km horizontal resolution) with many more members (Bouttier et al. 2016). Hermoso et al. (2020) recently
proposed a multiscale-consistent ensemble generation method that adapts to the scaling requirements of the "flow of the
day" by means of a seamless scale bred vector technique. This is especially suitable for Mediterranean cyclone prediction

applications due to the wide range of process scales involved. Indeed, such scale-adaptive ensemble generation techniques
produce large mesoscale perturbations during the initial baroclinic phase of cyclogenesis, while they are capable of exciting
convective scale modes at the mature and more diabatically influenced phases of the cyclone lifecycle. In the context of
representing uncertainties, Hermoso et al. (2021) tested model error representation through the application of stochastic
perturbations to physical parameterizations. Their application concerned a heavy precipitation event and showed that

perturbations to specific parameters within the microphysics parameterization outperform a multiphysics-based approach.
The positive impact of stochastic parameterizations for the forecast of extreme events in the region is attributable to the
generation of EPS spread at small scales, which is a rather difficult and rarely sampled source of important uncertainties for
these events. This also simplifies the technical management of model data in forecasting applications, since there is no need
to maintain multiple parameterization codes and data streamflows as in other EPSs based on multiphysics or multi-model

approaches.

## 3.3 Use of observations and data assimilation

Insufficient routine observations for data assimilation requires the adoption of strategies that account for regional
specificities, such as severe coastal impact of intense convective systems initiated over the sea within cyclones and
interactions between the airflow and the mountains. A systematic investigation of the forecast challenges related to

Mediterranean cyclones was performed during MEDEX through the production of one of the first databases of
Mediterranean cyclone tracks (Campins et al. 2011; Picornell et al. 2011). In this framework, high impact weather prediction
in the Mediterranean was improved as a direct result of advancements in understanding the most important
(thermo-)dynamical processes in Mediterranean cyclones and of the onset of observational targeting strategies (Jansa et al.,
2014). Running a Data Targeting System (DTS) was one of the emblematic coordinated milestones of MEDEX, which

aimed at identifying sensitive areas where improvement in type and number of observations are likely to reduce forecast
errors.

However, Campins et al. (2013) found that the benefits of DTS on operational analyses and forecasts is very variable and
highly dependent on the relative location of the sensitive areas with respect to the distribution of the operational network of

observations, as well as on the particular predictability characteristics of the case under analysis. Similarly, Garcies and
Homar (2014) verified the skill of the sensitivity products used in the MEDEX-DTS campaign. They found that objective





sensitivity fields fail to recognize the best location where extra observational means should be deployed. Indeed, observing system experiments run in that campaign revealed that pseudo-soundings in areas highlighted by objective sensitivity calculation methods have negligible influence on the forecast of intense cyclones, even weaker than pseudo-soundings in operational sounding sites, allegedly unrelated to the dynamics of the day. Therefore, the challenge of identifying useful targeting strategies to improve Mediterranean cyclone prediction remains open. It is noteworthy that, despite the limitations imposed by their linear character, sensitivity calculation methods allow to derive significant sensitivity information in climatic mode.Indeed, Garcies and Homar (2009, 2010) point towards the European Atlantic, the "Mediterranean Storm Track" from the Atlantic to the Ionian Sea and the north-western African lands as the areas showing the highest sensitivity for the 24-48 h forecast of intense Mediterranean cyclones.

Regarding the scarcity of observations, Carrió and Homar (2016) proposed the use of Ensemble Kalman Filters (EnKFs) for calculating an ensemble of temporally and spatially evolving forecast errors and thus to derive a dynamical background error covariance matrix. Modern EnKFs typically employ "continuous cycling" that unifies ensemble forecasting and analysis steps. This seamless integration renders initial ensembles dynamically consistent in contrast to initial ensembles generated by downscaling external models. This cycling configuration shows promising applications to forecasting Mediterranean cyclones since it can effectively transport observational information from land and coastal areas towards less densely observed regions, significantly improving the representation of maritime cyclogenetic environments. In fact, ensemble data assimilation methods based on Kalman filtering, combined with stochastic physics and the comprehensive ingestion of remote sensing measurements, is the cutting edge framework to tackle the challenge of efficient forecasting of Mediterranean cyclones and their impacts. Carrió and Homar (2016) and Carrió et al. (2019) show that improved analysis of key cyclogenetic fields such as low levels temperature, specific humidity, or wind improves the 12-18 h prediction of small scale aspects of the cyclone structure and thermodynamic settings. This, in turn, contributes to better forecasting high-impact weather associated with cyclones. In this context, Lagouvardos et al. (2013) have highlighted the positive impact of lightning assimilation in the representation of the precipitation field of a Mediterranean cyclone that affected southern France. However, the assimilation of remote sensing data, such as radar and satellite radiances or motion vectors, has proven very challenging for a distinctly multiscale problem such as the Mediterranean cyclogenesis and its impacts. In fact, these remote sensing measurements are nonlinearly related to the model state, which itself is a product of highly nonlinear dynamical–microphysical processes (Aksoy et al., 2010).

## 3.4 New modelling opportunities and challenges

Continuous developments in model physics and increase in computing power have opened new opportunities in numerical weather prediction, which have been applied to several case studies of Mediterranean cyclones. A straightforward improvement is the increase in resolution, as global forecasting systems typically cannot fully resolve the structure and intensity of compact vortices such as medicanes. During HyMeX, the flagship model contribution was the AROME-WMED





(Fourrié 2015, 2019), which featured a kilometric resolution analysis over the Western Mediterranean, covering the special observation period (SOP; Ducrocq et al., 2014) at an unprecedented resolution for the region, and with very significant improvements in moisture representation from Navigation Satellite System data (Fourrié, 2020). As an example of the usefulness of high resolution simulations, Carrió et al. (2020) analysed the genesis of a small secondary cyclone, embedded within the broader cyclonic circulation of a primary cyclone that travelled along the northern half of the Western

Mediterranean and produced heavy rainfall over Italy (Flaounas et al., 2016). Figure 6 shows the development of this secondary cyclone and its relationship to convection. Carrió et al. (2020) used piecewise PV inversion to further show that diabatic heating within the secondary cyclone was crucial to intensify the primary cyclone. The high resolution outputs of AROME-WMED were the key to diagnose the intense winds that were associated with the secondary cyclone and that affected the Balearic Islands.


Horizontal grid spacings of a few km or less, which alleviate the need for a parameterization of deep convection ("convection-permitting" or "storm-resolving"), have been widely used for quantitative precipitation forecasts in the Mediterranean region. For instance, Cioni et al. (2018) found explicit convection necessary to capture the track, intensity and thermal structure of the November 2014 medicane. They also observed further improvement by adding a nested sub-km

scale domain. The sensitivity to the representation of convection was attributed to the crucial interaction between convectively-produced PV at low levels and a PV streamer aloft. Although this result appears promising, a systematic gain from km-scale resolution has not been generally demonstrated for cyclones yet. Besides, improvements of physical parameterizations are needed, in particular for microphysics and turbulence, since they remain two major sources of uncertainty. Embedded convection in WCBs is sensitive to the representation of subgrid turbulence, which has been shown

to be underestimated in convective clouds (Verrelle et al., 2017, Strauss et al, 2019). Furthermore, the PV development in WCBs is strongly modified by the latent heating associated with microphysical processes (Joos and Wernli, 2012). Changes in their parameterization can thus lead to differences in the position of the WCB outflow (Joos and Forbes, 2016, Mazoyer et al, 2021). Thus, their accurate representation is important both locally for predicting precipitation, and non-locally, when latent heating in North Atlantic WCBs potentially affects the prediction of the Mediterranean cyclone downstream (Raveh-

Rubin and Flaounas, 2017).

Recent years have also seen the emergence of coupled modelling systems for the atmosphere, ocean and/or waves. Due to the analogy with tropical cyclones, for which ocean-atmosphere coupling reduces SST due to upwelling and weakens the cyclone, research has mainly focused on medicane case studies so far. Low sensitivity to coupling has been found in

climatological studies, partly attributed to the currently relatively-coarse resolution of regional climate models (Flaounas et al. 2018; Gaertner et al. 2018). Such low sensitivity of cyclone dynamics was however confirmed by recent tests applied to case studies. For instance, the November 2011 medicane exhibited weak dependence on ocean-wave-atmosphere coupling with 5 km grid-spacing, while model uncertainty was dominated by initial conditions (Ricchi et al. 2017). Similarly, the





track and intensity of the November 2014 medicane were weakly impacted by ocean-atmosphere coupling, even at km-scale

resolution; this was ascribed to surface cooling one order of magnitude smaller than that for tropical cyclones (Bouin and Lebeaupin Brossier 2020). For the September 2018 medicane, Varlas et al. (2020) showed that coupling ocean and atmosphere models with ~10 km grid spacing cooled the ocean surface by up to 2–3°C with respect to the model initial conditions. Additional 3-way coupling with a wave model reduced 10-meter wind speed by a maximum of 2-3 m/s during the mature stage of the cyclone. Finally, similar conclusions were reached by Stathopoulos et al. (2020) who showed that the

dynamics of three medicanes (Trixie, October 2016; Numa, November 2017; Zorbas, September 2018) were rather insensitive to different modeling approaches on SST (including coupled and stand alone atmospheric, high resolution simulations). Despite the weak impact of explicitly resolved air-sea interactions on cyclone dynamics, more research efforts are needed to elucidate the potential benefits brought to dedicated forecasting systems, e.g. to coastal oceanography. In addition, further research is needed to expand the impact of explicitly resolved air-sea interactions to cyclone cases other

than medicanes.

## 4 Climatological perspective of Mediterranean cyclones

Mediterranean cyclones represent a well distinct element of the global climate system. Several studies between the 1950's and 1960's brought to light the high frequency of cyclogenesis in the Mediterranean (Pisarski, 1955; Petterssen, 1956; HMSO, 1962; Radinović, 1965). Notably, the analysis of Petterssen indicated the Mediterranean as the most active

cyclogenetic area in the Northern Hemisphere in winter. In later years, the effort to unveil the climatological aspects of Mediterranean cyclones continued through the use of gridded datasets and objective identification methods. This section provides an overview of the main results of these efforts, focusing on the climatology of cyclone tracks, the different cyclone types that develop in the region and their future trends.

### 4.1 Spatial variability and seasonal cycle of Mediterranean cyclone tracks

Alpert et al. (1990) performed one of the first studies focusing on the climatology of Mediterranean cyclones. The authors used a 5-year (1983-1987) climatological dataset of ECMWF operational analyses with a grid spacing of 2.5°x2.5° and a time interval of 12 hours. The monthly variations of cyclonic tracks in the entire Mediterranean showed a pronounced seasonal cycle. Cyclone occurrences are more frequent in winter, with tracks located along the northern and central part of the Mediterranean basin, while in summer, cyclone occurrences become frequent over North Africa. In later years, a high

number of studies were devoted to produce a climatology of cyclones, with various results on the spatial and temporal variability of their tracks (e.g. Trigo et al. 1999; Maheras et al. 2001; Campins et al. 2000; Picornell et al. 2001; Bartholy et al. 2009; Flocas et al. 2010; Campins et al. 2011). As an example, Fig. 7 (top panel) shows the spatial distribution and the seasonal cycle of cyclones, but only for the most intense systems (about 30 cyclones per year), defined by the geostrophic circulation in their centres exceeding a threshold of $7\times10^7$ m$^2$ s$^{-1}$ (Homar et al. 2006). Figure 7 (bottom panel) also shows the





seasonal cycle of notable case studies, distinguished in the framework of MEDEX for their importance in terms of impacts

and intensity. Most of the intense cyclones in Fig. 7 are formed in the northwest Mediterranean and more precisely within

the Tyrrhenian and Adriatic Sea, whereas cyclogenesis is frequent in the vicinity of the Alps (see Sect. 2.4). Another branch

of cyclone tracks departs close to the Atlas mountains and enters the central part of the basin from south-west, while other

areas of high track density are located in the Ionian, Aegean and Black Seas, and Levantine basins close to Cyprus.


Regarding the seasonal cycle of intense cyclones, there is a substantial agreement among studies that cyclogenesis is more

frequent in winter and exhibits a flat minimum in summer (Campins et al. 2011; Lionello et al. 2016; Flaounas et al. 2018).

More precisely, winter cyclones that develop in the Gulf of Genoa are often deep systems, and, once formed, they travel

southeastwards, affecting with high-impact weather the Alpine region, the Italian Peninsula, the Adriatic sea, and the central

and eastern part of Mediterranean Sea (Horvath et al., 2008; Ulbrich et al., 2012; Reale et Lionello, 2013). Cyclones around

Cyprus, usually called Cyprus Lows (e.g. Alpert et al. 1990; Flocas et al., 2010; Ulbrich et al., 2012), play a fundamental

role in shaping the variability of precipitation in the eastern Mediterranean basin (Goldreich 2003; Saaroni et al., 2010;

Hochman et al., 2019). Autumn and spring are transitional periods for cyclogenesis frequency, sharing to some extent similar

characteristics with the climatological spatial variability of winter. However, in spring, cyclones occurrence shows a distinct

local maximum in Northwest Africa, at the leeward side of the Atlas mountains corresponds to a favourite area of spring

cyclogenesis, from which cyclones travel eastwards, along the Northern Africa coastlines, towards the Middle East (Egger et

al. 1995; Thorncroft and Flocas 1997). Finally, in summer, several shallow cyclones may occur close to Cyprus, in the

Sahara, the Iberian Peninsula, the Gulf of Cadiz, the Gulf of Genoa and the Algerian Sea. Although identified as cyclones,

these systems do not correspond to organised, mesoscale wind vortices. Such thermal shallow depressions, confined to the

atmospheric boundary layer, should not be confused with active and deeper cyclones. Indeed, the summer position of the

Polar jet renders upper tropospheric systems less likely to intrude on the Mediterranean and trigger cyclogenesis (Flaounas et

al. 2018).

The actual number of Mediterranean cyclones varies significantly among climatological studies from hundreds to even

thousands of occurrences per year. Such high uncertainty on the number of tracks in  the Mediterranean is related to the

sensitivity of the cyclone tracking methods, the criteria used to identify cyclones and the properties of the input dataset (e.g.

grid spacing, temporal interval of fields etc). It is thus not surprising that the actual number of Mediterranean cyclones and

the  characteristics of their tracks are highly dependent on the cyclone tracking method (Neu et al. 2013). For instance,

Lionello et al. (2016) and Flaounas et al. (2018) found substantial differences in the number of cyclone tracks when

comparing the outputs of different cyclone tracking methods. However, consensus on the spatial and seasonal variability of

the tracks is high considering intense Mediterranean cyclones. On the other hand, results for the relatively weak cyclones

present high variability, even their seasonal cycles of occurrence (Campins et al. 2011; Lionello et al. 2016). Consequently,

new results on the climatological analysis of Mediterranean cyclone tracks should always be discussed along with the





physical and technical criteria of the various methods used to identify and track cyclone centres. In addition, it is imperative
to compare new results  with those of the rich bibliography on the climatology of Mediterranean cyclone tracks. This is a
crucial issue not only for the effective advancement of the state-of-the-art, but also for the community to contextualise the
existing climatological results to the dynamics and physical aspects of the included cyclones.

## 4.2 Subtypes of Mediterranean cyclones

Several studies identified a large variety of cyclone subtypes. The criteria to identify a cyclone subtype are of different
nature: geographical origin, cyclogenesis processes, deepening rate etc. In this section, we discuss the climatological aspects
of several cyclone subtypes that were specifically addressed in past and recent studies.

*Cyclones forming over the Atlantic Ocean:* It is not uncommon for cyclones forming over the Atlantic to intrude into the
Mediterranean region and cause substantial damages (Nissen et al. 2010). Such an example is storm Klaus that intruded the
Mediterranean in January 2009 (Liberato et al., 2011). In a most recent study, Lionello et al. (2016) used 14 different
cyclone tracking methods from the IMILAST dataset (Neu et al., 2013) and showed that the Mediterranean cyclones that
form within the Mediterranean represent 61-85% of the total Mediterranean tracks.

*Medicanes:* The dynamics of known medicane cases have been already discussed in section 2.5. Regarding their
climatology, several studies focused on tracking medicanes in multi-year datasets. However, lack of physical definition of
medicanes produced different climatologies. The criteria used varied from subjective identification of tropical-like
characteristics (Tous et al., 2013) to the application of Hart diagrams and wind thresholds in high resolution simulations (e.g.
Cavicchia et al. 2014a). Recent studies used dynamical downscaling of ERA-Interim reanalysis (Gaertner et al., 2018;
Ragone et al., 2018) and ERA5 reanalysis (Zhang et al., 2020; de la Vara et al., 2021) to identify medicanes in a
climatological framework. Results showed rather large differences in their climatologies even among same model
simulations but using different parameterizations. In particular, the study of Gaertner et al. (2018) showed that medicane
climatologies were sensitive to the wind speed criteria and thus to the model horizontal resolutions, while air-sea coupled
models produced a seasonal shift to a winter frequency maximum compared to a maximum in autumn in uncoupled models.

*North African cyclones*: A high number of cyclones form on the leeward side of the Atlas mountains. These fast traveling
systems are known as desert depressions (Pedgley 1972) or "Sharav cyclones" (Winstanley 1972) and, as discussed in the
previous section, they present a clear climatological peak during spring time. Their formation over the arid areas of North
Africa makes Sharav cyclones to hold a low moisture content and to be associated with dust storms (Alpert and Ziv 1989;
Egger et al., 1995; Saaroni et al. 1998; Karam et al. 2010; Flaounas et al. 2015). Sharav cyclones are usually smaller and
propagate faster than Mediterranean winter cyclones (Alpert and Ziv 1989) and, due to their location, it is rather likely that
the subtropical jet plays an important role in the dynamics of these systems (Prezerakos et al., 2006).





*Explosive cyclones:* An explosive cyclone, or weather bomb, is characterized by a strong deepening over a relatively short time range. For the Mediterranean, Kouroutzoglou et al. (2011) suggested a deepening rate of 20 and 14 hPa in 24 hours for

northern and southern Mediterranean, respectively. A broad range of processes of the upper and lower troposphere have been associated with the surface rapid deepening, including baroclinic forcing of PV streamers and latent heat release due to convection (Kouroutzoglou et al. 2012; Reale et al. 2019). In the Mediterranean, explosive cyclogenesis is a rather rare maritime phenomenon with 5 to 6 occurrences per year. These occurrences follow the same climatology of intense cyclones, presenting a maximum from November to March and typically taking place within northern locations up to 40oN. More than

half (57%) of the cyclones that deepen explosively in the Mediterranean form within the region itself, with 43% in the western and 14% in the eastern Mediterranean (Kouroutzoglou et al. 2011; Reale et al. 2019). The rest of the systems that become explosive within the Mediterranean form outside the Mediterranean and they mainly intrude into the region from the northwest, as storm Klaus or Gong (Liberato, 2014).

*Vb Cyclones:* Vb cyclones (van Bebber, 1891) are described as low-pressure systems, with highest occurrence in spring (Messmer et al. 2015), propagating northeastwards from the western Mediterranean Sea to Central Europe, through northern Italy and leaving the Alpine ridge on the left. They are responsible for high-impact weather over the northern side of the Alpine range and Central Europe (Nissen et al. 2013). However, Vb cyclones are rather rare systems, constituting only 5% of all Central European cyclones, with about 2.3 Vb appearances per year according to Messmer et al. (2015). Regarding the

onset of an Vb event, the superposition of the polar jet stream and the subtropical jet stream over the western Mediterranean has been identified as a prominent feature (Hofstätter and Blöschl 2019).

*Secondary ('daughter') cyclones:* These systems refer to cyclones that form within the periphery of existing cyclones and have received separate interest. Their climatological characteristics have been investigated by Ziv et al. (2015) who found

that the majority of Mediterranean cyclones are 'daughters' of 'parent' cyclones that are located in the Mediterranean itself. Parents of Genoa lows are mostly located east of their daughters, implying that cyclogenesis takes place within the northerly flow that crosses the Alps. Parents of Aegean Sea lows are located mainly west of the daughters, implying that the cyclogenesis takes place within the southerly flow ahead of the parent cyclones, i.e. within their warm sector or at the warm front of the parents. Further analysis by Saaroni et al. (2017) showed that over half of 'daughters' formed on the frontal

system of the 'parent', one third on a separate frontal system and about 10% within the warm sector of the 'parent' cyclone.

## 4.3 Large-scale circulation and Mediterranean cyclones

In the North Hemisphere circulation, especially for winter, teleconnections account for a significant portion of the atmospheric variability (e.g. Nigam and Baxter, 2015). The most prominent teleconnection pattern, in terms of influence on the Euro-Mediterranean climate, is the North Atlantic Oscillation (NAO), which has a wide spectrum of scales ranging from





560 the intraseasonal to the decadal timescales. Nevertheless, different atmospheric oscillations coexist with each other, namely the Arctic Oscillation (AO), the Eastern Atlantic (EA), Scandinavian (SCAN) and East Atlantic-West Russia (EAWR) patterns. These evolve and interact simultaneously, thus increasing the complexity of their impacts on the regional climate (e.g. Krichak and Alpert 2005a,b; Trigo et al. 2008).

565 It is largely accepted that NAO and the Atlantic storm track are closely related (e.g. Hurrell and van Loon 1997; Woolings and Blackburn 2012): positive NAO phases are related to northward shifted and intensified storm tracks, while negative NAO phases are related to more cyclones in the western Mediterranean and southern Europe (Raible, 2007; Pinto et al. 2009). The latter was confirmed by Nissen et al. (2010), who found more cyclones crossing the Mediterranean during a negative NAO phase (approx. 20%) compared to its positive phase. On the other hand, they found that positive NAO and

570 EAWR are both associated with strong wind events in the eastern Mediterranean, while the positive EAWR phase is connected with a decrease of wind tracks in the central Mediterranean. More recently, Hofstätter and Blöschl (2019) found that Vb events tend to occur during negative NAO phases, while NAO and AO appear to be synchronized with Vb clustering (i.e. successive Vb cyclones), which is more frequent when both NAO and AO are negative. On the other hand, Vb cyclones were less frequent from 1988 to 1997 when a sustained positive phase of both NAO and AO prevailed.

575 **4.4 Current and future trends of Mediterranean cyclones**

The Mediterranean is considered as particularly vulnerable to climate change (e.g. Giorgi, 2006; Solomon et al. 2007). In these regards, cyclones play a crucial role in climate variability and its extremes. Considering the period 1979–2008, Lionello et al. (2016) used an ensemble of cyclone tracking methods to show that there is no clear interannual trend of cyclones occurrence. On the other hand, Flocas et al. (2010) performed a 40-yr (1961-2000) analysis in the eastern

580 Mediterranean and showed a negative trend, especially regarding winter cyclones. A negative interannual trend is also consistent with the results of Trigo et al. (2000), but only concerning the most intense systems.

Several studies used global and regional models to analyse the future trends of cyclones occurrences and intensity, often reaching contrasting results (Lionello et al. 2002; Bengtsson et al. 2006; Lionello et Giorgi, 2007; Raible et al, 2010; Zappa

585 et al. 2013; Nissen et al. 2014; Zappa et al, 2015; Hochman et al., 2018; Reale et al. 2021). One among the first studies was performed by Lionello et al. (2002) who used the ECHAM-4 global model, forced by the doubled $CO_2$ emissions experiment, finding a diminished overall cyclone occurrence. Later, Lionello and Giorgi (2007) used a regional climate model under the SRES emission scenarios A2 and B2 to show a weaker rate of cyclones occurrence in the East Mediterranean for the period 2071-2100 (compared to the period 1961-1990) and a stronger activity in the north-west part of

590 region. Raible et al. (2010) used the ECHAM model under the emission scenario A2 and for the period 2071-2100 to show a decrease of 10% in cyclones occurrence in the West Mediterranean (with respect to the period 1961-1990). However, no significant changes were found in the East Mediterranean but a slight decrease in the cyclone intensity in the entire region.





Modifications were attributed to changes in baroclinicity, static stability, transformation from eddy kinetic energy to kinetic energy of the mean flow and stationary wave activity particularly in the West Mediterranean and along the coastline of North Africa. The multimodel analysis of Nissen et al. (2014) indicated that the cyclone decrease, simulated under climate change conditions (SRES A2) over most of the Mediterranean, can be associated with a shift of NAO to a more positive state, which can explain 10-50% of the negative cyclone trend over the Gulf of Lion. More recently, Zappa et al. (2015) used an ensemble of 17 CMIP5 models under emission scenario RCP8.5 and found an overall reduction of cyclone occurrences in the period 2082-2099 (compared to 1976-2005), which, in turn, is responsible for Mediterranean precipitation reduction in winter. This result comes in accordance with the recent study of Hochman et al. (2018) who showed a reduction by 35% of cyclone occurrence in the East Mediterranean towards the end of the century using 8 CMIP5 models under the RCP8.5 scenario. Finally, Reale et al. (2021) used the Med-CORDEX ensemble of seven regional coupled system models. Results showed an overall reduction of the number and intensity of intense Mediterranean cyclones at the end of the 21st century under emission scenario RCP8.5. In addition, the authors found an increase (decrease) in cyclone-related wind intensity and precipitation in the Central (Southeast) part of the region. However, modelled climatologies presented a non-negligeable spread in the future trends of cyclone deepening rates, seasonality, cyclone-related wind and precipitation patterns, especially in the Ionian Sea and Levantine basin, highlighting the necessity for ensemble approaches and model improvements.

Regarding cyclones intensity, Gaertner et al. (2007) used an ensemble of regional climate models (RCM) to dynamically downscale a single GCM. They found an intensity increase under the A2 emission scenario for early-autumn cyclones, although the uncertainty among RCMs was large. At least for the most sensitive model, the future intensity increase in cyclones was associated with the development of clear tropical characteristics. After this study, a rather general agreement has been reached about two main aspects of the future evolution of medicanes: a frequency decrease in parallel to an intensity increase. Very different climate model setups have been used in this respect: a single GCM-RCM pair (Walsh et al., 2014; Cavicchia et al., 2014b), one high-resolution atmospheric GCM (Tous et al., 2016) or multiple GCM-RCM pairs (Romera et al., 2017). Romera et al. (2017) used a large ensemble of models and of long multidecadal reference and future periods (1951-2000, 2051-2100). They confirmed this trend considering a higher number of medicanes, though an important inter-model uncertainty remained, as a few GCM-RCM pairs showed a decreasing intensity trend. More recently, González-Alemán et al. (2019) and Gutiérrez-Fernández et al. (2021) used coupled RCM models to further investigate the effect of explicitly resolved air-sea interactions on medicanes future trends. Despite the possible impacts of a negative intensity feedback in coupled simulations due to the SST cooling effect of strong winds, the main future trends of medicanes were not different in these two studies.

The overall analysis on medicane physical characteristics and climatology is based on a rather small number of cyclones. In these regards, the application of a statistical-deterministic method to the study of medicanes by Romero and Emanuel (2013; 2017) strengthened the statistical robustness of the increasing intensity trend. However, their methods strongly relied on





medicanes sharing similar dynamics with tropical cyclones (as in group 2 in section 2.5). The overall agreement of different studies on the future increase of medicanes intensity has been largely attributed to an increase of SST in the Mediterranean basin. In future atmospheric conditions, higher SSTs are expected to enhance convection in contrast to the atmospheric

temperature increase (higher in the upper troposphere) which favours atmospheric stability. This contrasting effect on cyclone dynamics was investigated by Koseki et al. (2021) through the application of a pseudo-global warming approach (Schär et al. 1996) to medicane Rolf. Results showed that competing effects resulted in a moderate increase of the intensity of the medicane. Despite the relative agreement among the different studies on medicane future evolution, it is noteworthy that all studies applied different criteria for medicanes detection. As discussed in section 2.5, this is due to the absence of

commonly accepted medicanes definition. Consequently, more studies need to address future trends of medicane dynamics with regards to other implicated atmospheric processes, especially upper tropospheric forcing.

## 5 Mediterranean cyclones as major environmental risks

A grand majority of weather induced natural hazards in the Mediterranean is related to Mediterranean cyclones (Lionello et al. 2006) with severe socio-economic consequences. Indeed, in the natural hazards report of Munich Re at global scales for

2019, the Mediterranean region was pointed out because of the storms that produce flash floods and inundation. Such storms have recurrently affected Italy, France, and Spain, inflicting damages and billionaire losses (Munich Re, 2020). Mediterranean cyclones with strong circulation, embedded convection and frontal structures, produce high impact weather including heavy rainfall and flooding, thunderstorms with intense lightning activity, windstorms and dust storms, tornadoes, storm surges and landslides, including compounding effects.

### 645  5.1 Heavy precipitation and relevant impacts

The important role of cyclones in provoking the majority of heavy rainfall was established by Jansá et al. (2001) in the context of MEDEX. The authors used observations and cyclone databases to show that 90% of heavy rainfall events in the western Mediterranean were indeed associated with cyclones of diverse intensity. More recently, Flaounas et al. (2018) showed that weak and intense Mediterranean cyclones are both capable of producing high precipitation amounts. On the

other hand, when considering rainfall averages and cyclone intensities, a linear relationship was found between rainfall and cyclones intensity. Regardless of their intensity, cyclones have been shown to play a dominant role in regional precipitation with contributions exceeding 70% of the annual total (Hawcroft et al. 2012; Pfahl and Wernli 2012). Concerning extremes, Pfahl et al. (2014) and Flaounas et al. (2016) analysed climatological datasets from reanalysis and regional climate models to show that, depending on the region, intense cyclones may contribute 60% and up to more than 90% to regional extreme

rainfall. The processes that produce heavy precipitation and relevant extremes might differ from case to case. In a global analysis Pfahl et al. (2014) showed that about 40 to 50% of the regional extremes can be attributed solely to WCBs in the Mediterranean (feature 5 in Fig. 2), while Flaounas et al. (2018) showed that WCBs typically produce more than twice the



precipitation amounts related to deep convection (features 6, 7 and 8 in Fig. 2). These two different processes of precipitation
production were analysed in two contrasting case studies that took place during the SOP1 campaign of HyMeX in October
2012 and were both located in the West Mediterranean, close to the Alps (Flaounas et al. 2016). Figure 8 shows similar daily
precipitation amounts in different areas of the West Mediterranean. However, high precipitation amounts in the first cyclone
were mainly produced by deep convection, depicted in Figs 8a, 8b and 8c by intense lightning activity and diagnostics from
satellite retrievals. On the other hand, deep convection in the second case was scarce (Fig. 8d) since high precipitation
amounts were mainly due to stratiform rainfall, produced by WCBs.


Heavy precipitation in the region requires large amounts of water vapour at low and mid-tropospheric levels, favouring
atmospheric instability and the deepening of systems leading to intense rainfall events (Reale and Lionello, 2013; Khodayar
et al. 2018b). Using composite fields based on mesoscale reanalyses, Ricard et al. (2012) showed that heavy precipitation
events over Southern France are associated with instability and moisture located upstream over the Mediterranean Sea and
transported by a low-level jet constrained both by the synoptic pattern and the local orographic effects. Khodayar et al.
(2018a) further demonstrated, using high-resolution seasonal simulations in the western Mediterranean, that the later
association also holds for other heavy precipitation hotspots in the western Mediterranean such as northern and southern
Italy, East Spain and north Africa. Additionally, they  pointed out that the significant increase in atmospheric moisture and
instability prior to heavy precipitation occurrence could be traced back to at least 6–24 h before the initiation stage for the
majority of events. The Mediterranean Sea, but also the tropical and extratropical Atlantic Ocean and tropical Africa have
been identified as potential sources of moisture (Winschall et al., 2014; Chazette et al., 2016; Lee et al., 2017; Raveh-Rubin
and Wernli, 2016; Duffourg et al., 2018), whose transport may be organized in atmospheric rivers (Krichak et al., 2016). In a
recent study, Davolio et al. (2020) showed how an explosive deepening Mediterranean cyclone steered an atmospheric river
towards the Alpine slope (Fig. 9) where one of the most severe precipitation events of the last century in Italy occurred
(Giovannini et al., 2021). The computation of the atmospheric water budget over the Mediterranean basin showed that the
local contribution of evaporation from the sea was much less important than the moisture transported from the tropical
Atlantic, through Africa. Flaounas et al. (2019) also quantified the water vapour amount feeding Mediterranean cyclones,
through a Lagrangian approach applied to 100 cyclone events. Results showed a higher contribution from the Mediterranean
Sea, while the Atlantic Ocean appeared as a secondary, but nevertheless important source for water vapour. Similar results
were reached by Raveh-Rubin and Wernli (2016) who analysed the water sources for five selected cases of intense
Mediterranean cyclones that produced high impact weather. Finally, Flaounas et al. (2019) analysed cyclones with respect to
their precipitation efficiency, defined as the ratio of precipitation to integrated rainwater in the atmospheric column.
Precipitation efficiency was shown to be systematically higher over land than over maritime areas, suggesting that the
relationship between cyclones intensity and precipitation is further perplexed by the underlying surface.






In the framework of MEDEX, societal impact research has focused on flood events, populating the first database of its kind for the region. This database has been the basis for other relevant works in the framework of HyMeX, which resulted in the FLOODHYMEX database (Llasat et al. 2013; Papagiannaki et al. 2013). This effort has been recently updated and extended to a Mediterranean Flood Fatalities (MEFF) database spanning from 1980 to 2015 (Petrucci et al., 2019). Vinet (2019) and

Petrucci et al. (2019), based on the MEFF database and its extended version EUFF (European Flood Fatalities) covering the period 1980-2018, have shown that the flood-related mortality mainly occurs during early autumn in the western Mediterranean and during late autumn and winter in the East Mediterranean. This seasonal preference of flood-induced fatalities is correlated with the climatological patterns of heavy precipitation and implicitly with Mediterranean cyclone occurrences (Raveh-Rubin and Wernli, 2015). In fact, most of the floods, resulting from torrential rain in the region, both in

the West and East Mediterranean, are associated with organized convective systems or orographic precipitation due to synoptic scale forcing rather than intense isolated convective cells. In fact, Michaelides et al. (2017) showed that most of the flash floods in the Mediterranean are related to heavy precipitation produced by relatively weak cyclones but with long-lasting embedded mesoscale convective systems that interact with the topography. From the provided non-exhaustive list of catastrophic floods that spread across the Mediterranean it can be inferred that most floods -if not all them- were related to

the presence of a cyclone. However, a quantitative attribution of the role of Mediterranean cyclones to flood occurrence is not yet performed. In this context, intense or long-lasting rainfall is also considered as a frequent factor for landslides. Rainfall induced landslides have been reported in the literature across the Mediterranean countries (Trigo et al. 2005; Gariano et al., 2015; Polemio and Petrucci, 2010); however, to our knowledge no direct attribution of the triggering of the observed landslides to Mediterranean cyclone activity has been yet published.


Finally, a relatively under-addressed issue of convection-related risks concerns the contribution of cyclones to Mediterranean electrical activity. Galanaki et al. (2016) quantified this contribution combining lightning observations and reanalysis. Results showed higher contributions in autumn and winter, i.e. when intense cyclogenesis presents the highest rate of occurrence, accounting for 5-30% of the observed lightning over the Mediterranean Sea. These percentages are much higher

in winter, especially in northern Africa where cyclone contribution to lightning occurrence reaches up to 60%. It is however noteworthy that the study of Galanaki et al. (2016) was limited to intense cyclones, while lightning activity was only attributed to cyclone centres, excluding e.g. fronts. Therefore, more studies need to address this issue with more explicit methods that combine identification of frontal structures, cyclone tracks and ensemble of lightning observations.

## 5.2 Wind-induced risks

### 5.2.1 Windstorms

Many of the strong winds observed in the Mediterranean have been traditionally considered to belong to the category of local winds, like Mistral, Tramontana, or Sirocco. However, several of these winds are provoked by Mediterranean cyclones





or cyclones play a primary role in their intensity. For instance, the intensity of the Mistral winds depends on Genoa cyclones and their upper-tropospheric PV anomaly (Givon et al., 2021). In another example, the presence of a low pressure over the

Tyrrhenian Sea was responsible for an exceptional Sirocco storm in the Adriatic basin and eastern Alps (Giovannini et al., 2021). Regardless of their relationship to local winds, Mediterranean cyclones are directly related to windstorms, either by enhancing local winds or due to their cyclonic circulation. Nissen et al. (2010) was among the first to perform a climatological analysis of windstorm tracks over the Mediterranean and found that cyclones are responsible for the quasi-total of wind extremes. In fact, 42% of these cyclones formed within the region, while the rest intruded on the Mediterranean

from the Atlantic Ocean. Recently, the environmental conditions associated with extratropical cyclones responsible for tornadoes in Italy have been analysed (Tochimoto et al., 2021), showing that tornadoes concentrate mainly in the warm sector, and, secondarily, along the cold front.

### 5.2.2 Storm surges and sea waves

A direct risk of cyclone-induced windstorms is the formation of storm surges and the consequent coastal flooding. In fact,

the role of cyclones in raising the sea level in coastal regions is added to background conditions. These conditions include changes in the surface heat flux, the advection of heat by currents, the seasonal changes and climatic variations of the sea level, tides, infragravity waves and wave setup due to the dissipation of waves reaching the nearshore area. It is within this complex scheme of interacting processes that cyclones may produce extreme waves that, combined with heavy rainfall, can dramatically impact coastal areas with flooding and erosion. For instance, the city of Venice in Northern Italy is particularly

vulnerable to flooding due to storm surges (Cavaleri et al., 2019; Lionello et al., 2019); the most intense one in the recent past was associated with a small-scale cyclone travelling northward along the northern Adriatic basin (Ferrarin et al., 2021). The recent medicane Ianos (September 2020) was responsible for storm surges in Greece due to sustained winds of ~40 m/s (e.g. Smart et al. 2020; Lagouvardos et al. 2021), while Patlakas et al. (2020) showed a substantial contribution of medicanes to high sea waves and wind speeds. As an example, Fig. 10 shows, during the lifetime of Qendresa (November 2014), the

time duration in which 925 hPa wind speed and wave height exceeded the thresholds of 17.2 m/s and 4 m, respectively. The combination of high wind speed, significant sea waves and consequent storm surges, along with heavy precipitation, suggest increased risk for compound events, especially for compound flooding (Bevacqua et al. 2019).

Few studies have been performed on Mediterranean storm surges, mostly focusing on variability and trends under current

and future climate (Marcos et al. 2011; Androulidakis et al. 2015). In particular, Lionello et al. (2019) used model hindcast to show a link between cyclones intruding into the region from the Atlantic and positive sea level anomalies along the Western Mediterranean coastlines. In contrast, high sea levels in the Southeast and Northwest parts of the basin are associated with cyclones that originate from the western basin, while high sea levels near North African and East basin coastlines are affected by cyclones formed over Northern Africa and Levantine basin. In a recent study, Tadesse et al. (2020)

used data-driven modeling to quantify the relationship between the storm-surges and relevant predictors such as the wind





speed and mean sea level pressure at global scale. Storm surges strongly rely on drag coefficient and thus on the surface wind speed. However, there is a considerable lack of in-situ observations that renders the understanding of storm surges an open question, especially when wind speeds are higher than 30 m/s, in these cases the theoretical framework is not well known. Moreover, storm surges are impacted by swell propagation across different sectors of the cyclone and indirectly by

the rain that affects the development of the wind waves. Currently, little is known about these mechanisms in the Mediterranean region.

### 5.2.3 Dust transport and episodes of particulate matter

Several studies have assessed the relationship between mineral dust mobilization, transport and synoptic-to-global-scale weather systems. Satellite data and model simulations have identified baroclinicity and cyclogenesis processes as significant

meteorological patterns that result in dust uplift due to the strong surface wind fields (Schepanski et al., 2009). In most cases Mediterranean cyclones can develop close to dust sources, with their core being either over the African continent or further north in the Mediterranean Sea (Schepanski and Knippertz, 2011). Dayan et al. (2008) used 37 years of visibility observations correlated with PM10 dust concentrations to examine the annual and inter-annual occurrences of dust events over Israel. They reported a significant correlation between dust and cyclonic activity in Eastern Mediterranean. Kalkstein et

al. (2020) further demonstrate the link between the highest PM10 concentrations in Israel and intense Cyprus Lows. Using reanalysis and satellite retrievals of aerosol optical depth, Flaounas et al. (2015) found that cyclones contribute to 10-25% of the total number of days with dust over the eastern Mediterranean while this percentage rises to 30-70% when considering only extreme dust events. Categorizing Saharan dust events into synoptic meteorological types, Varga (2020) reported that well-developed Mediterranean cyclones drive almost a quarter of these, mobilizing and transporting dust through the

southerly warm advection ahead of the cold front of the eastward-moving low-pressure systems. As an exemplary case, Figure 11 shows satellite observations of a severe dust transport event due to a cyclone in March 2016, analysed by Rizza et al. (2018a). The relationship between the cold front and dust transport is clearly visible in the satellite pictures of Fig. 11, which presents different time instances of a major dust transport event due to a cyclone in March 2016. Close to the time of the cyclone's mature stage (Fig. 11b), the cyclone centre is located to the north of Tunisia, where cloud coverage (brown

colors) is wrapping cyclonically. In parallel, the cold front forms a comma shaped cloud coverage (brown colours), organised along a meridional direction over the Ionian and Adriatic Sea preceded by an also meridionally oriented filament of dust (pink colours). Clearly dust is uptaken during the cyclone formation over North Africa (Fig. 11a) and continuously interacts with clouds even in subsequent times, after the mature stage (Fig. 11c). The high contribution of cyclones to dust events was also confirmed in a series of case studies (Varga et al., 2014; Fiedler et al., 2013; Schepanski et al., 2009).

Finally, cyclones were also identified for facilitating the long-range transport of giant mineral dust particles (>75 μm in diameter). This has an important effect on the radiation balance of the atmosphere, cloud and precipitation processes and the ocean carbon cycle (Ryder et al., 2013; Van der Jagt et al., 2018). The assessment of the size distribution of mineral dust particles under cyclonic circulation conditions could prove an important, currently under-addressed, research objective.





## 6 Open questions and research perspectives in the broader field of Mediterranean cyclones

This article has presented the community's efforts over the last decades to advance the field of Mediterranean cyclones. The first aspect that we reviewed, in section 2, focused on cyclone dynamics and in particular on the processes that play an important role in the development of cyclones. Upper tropospheric forcing, baroclinic instability in the presence of orography and diabatic processes are certainly key issues for cyclone dynamics. Acquiring insights into these processes is the fundamental step to better understand the issues implicated in forecasting cyclogenesis and relevant impacts. Therefore,

section 3 addressed the recent advancements in forecasting Mediterranean cyclogenesis, highlighting the importance of large-scale dynamics for a skilful prediction of cyclones, especially of RWB over the Atlantic Ocean. Furthermore, we discussed recent strategies for the probabilistic forecasting of cyclone impacts and the latest modelling efforts in this direction, including resolution issues and coupled modelling. Present and future cyclone tracks and frequency were discussed in section 4, along with the different cyclone subtypes met in the Mediterranean region. Finally, section 5 provided an

overview of high-impact weather events. In particular, we discussed cyclones' contribution to heavy precipitation events, windstorms, severe dust episodes, storm surges and high sea waves. The overall field of Mediterranean cyclones experienced substantial advances in the last decades, especially in late years in the framework of MEDEX and HyMeX international programmes. However several open questions are yet to be addressed by the community as highlighted in the following, in eight subsections, each devoted to a broad thematic.

### 6.1 Disentangling Mediterranean cyclone dynamics across spatial scales

The morphological characteristics of the PV streamers and cut-off lows, as well as their role in cyclones development when interacting with the subtropical jet, are still an open question. More research efforts need to further address the relationship between cyclones development and the amplitude of induced PV anomalies, their spatial extent and depth in the troposphere. The interaction of convection with upper tropospheric systems is also an underadressed issue. It is rather unclear how typical

it is for explosive cyclogenesis to take place due to a PV streamer triggering deep convection as described in Fita and Flaounas (2018), or for a cyclone to deepen abruptly due to the synergy of a PV streamer and preexisting convection as described in Chaboureau et al. (2012). The proper representation of the scales interplay by atmospheric models, e.g. between large-scale, upper tropospheric systems and organised convection, is crucial for the understanding of frontal structures and relevant airstreams. The latter is a relatively under addressed issue, with few studies performing systematic identification of

fronts and their role in cyclone dynamics and impacts. Late efforts on the identification of fronts in the Mediterranean (e.g. Bitsa et al. 2019) need to be further related to cyclonic systems and relevant impacts. Furthermore, the field suffers from the lack of dedicated, systematic analysis of mesoscale airstreams(WCBs, CCBs, DIs and sting jets). These are two main issues where the field of Mediterranean cyclones lags behind the one of mid-latitude storms developing over open oceans. Finally, a remaining issue deserving clarification is the PV-topography interaction in case of lee cyclogenesis, that is how the

secondary lee cyclone circulation feeds back and modifies the upper PV primary anomaly.





## 6.2 Identifying the specific characteristics of medicanes

Several issues on medicanes remain the subject of debate in the scientific community. Indeed, open questions on processes still concern the role of vertical wind shear (especially under the presence of superposed jets) and the potential of tropical transition taking place in the region. The latter includes the investigation of DI's implication to warm seclusion (e.g.,

Miglietta et al., 2021), the role of frontal structures, and the possible role of convection in sustaining alone the cyclonic circulation (Dafis et al., 2020). In addition to open questions on atmospheric processes, other subjects range from the criteria for the reliable definition of medicanes to their classification, both as a particular category and as a member of the broad category of subtropical cyclones, with the possibility of acquiring a tropical cyclone structure in a few cases. In this context, concepts such as WISHE, a cloudless "eye" and tropical transition should be used with caution when applied to medicanes.

Indeed, medicanes are still physically ill-defined systems and their dynamics present case-to-case variability. It is thus important for coordinated research to develop new diagnostic tools that take into account the unique dynamics and physical characteristics of Mediterranean cyclones.

## 6.3 Understanding coupled processes in Mediterranean cyclones

Including missing coupled processes in modelling approaches is an important step to advance our understanding of cyclone

dynamics. For instance, the role of dust and other particulate matter in modulating cyclone dynamics is still an open question, especially in the Mediterranean, a region that hosts abundant concentrations of aerosols (Lelieveld et al., 2002) and where cyclones are responsible for major events of dust transport (Flaounas et al., 2015; Rizza et al., 2017). In these regards, the direct effect of dust and other particulate matter on radiation and the indirect effect on microphysics is not yet systematically addressed in the context of cyclones development. In addition to dust, air-sea interactions are also an issue

that deserves more attention. Despite the limited effect of explicitly resolved air-sea interactions on certain medicane cases, it is still imperative to test the performance of coupled models to a wider range of cyclones. For instance, we need more studies on three-way coupling atmosphere-ocean-waves to understand the fine scale dynamics of cyclones and their consequent impacts on coastal areas. In the same context, aerosol-aware microphysics parameterizations, including sea salt emissions, are also important since they affect the intensity of cyclones and the consequent precipitation (Pravia-Sarabia et

al. 2021; Rizza et al. 2018b, 2021; Varlas et al. 2021). Such efforts will evolve to a full scale modelling framework that can address a wide range of coupled processes with substantial applications to numerical weather prediction. However, the use of coupled models is still relatively limited in the field of Mediterranean cyclone dynamics and has to be further promoted through interdisciplinary studies.

## 6.4 Solving resolution and parameterization issues for cyclone modelling

Due to the variety of processes that interact at different spatiotemporal scales and affect cyclone dynamics, the use of high resolution models is mandatory to advance our understanding and perform skillful forecasts of cyclones and their relevant





impacts. The complex geographical characteristics of the region and the relatively small size of Mediterranean cyclones would demand a considerable increase in resolution compared to current operational models for the area. Therefore, new studies should promote systematic analysis of cyclone dynamics using convection-permitting resolution. This would waive
inherent uncertainties in convection parameterizations and would provide considerable insights into the local impacts of cyclones. In addition, higher resolution would provide further insights into the extent of the impact of processes already resolved even in coarse resolutions. For instance, orography representation is still a source of uncertainty in the prediction of lee cyclones, since it limits the correct description of the mesoscale forcing. The spatial orographic scales that affect cyclone dynamics are still an open question and it is rather uncertain whether higher resolution is increasing the amplitude of
topographic-produced PV anomalies. Such processes might prove more important than the secondary role considered in past studies. In this context, new studies should re-examine results of already analysed cases, or further refine the conclusions by highlighting fine scale atmospheric processes. However, it is important to stress that waiving the use of convection parameterisation is still not assuring the reliable reproduction of latent heat and relevant forcing to cyclone dynamics. In fact, cyclone tracks and other cyclone characteristics such as intensity are highly dependent on model microphysics (e.g. Miglietta
et al., 2015; Pytharoulis et al. 2018) and thus higher resolutions and use of coupled models should advance in parallel to the representation of inherent uncertainties in physical parameterizations. In particular, further studies are necessary to assess the performance of two-moment versus one-moment microphysics schemes. Differences in the processes of vapor deposition on hydrometeors in cold and mixed-phase clouds can lead to differences in heating rate along the WCBs.Additionally, the representation of turbulent mixing has to be improved not only in the planetary boundary layer but also in the free
troposphere, as the strong wind gradients associated with cyclones can be a source of severe turbulence.

**6.5 Enhancing the use of observations and diagnostic tools**

In contrast to the wide availability of modelling techniques, the field of Mediterranean cyclone dynamics suffers from the lack of observations. Satellite data can provide useful information over the Mediterranean; for example, the wind products from the Aeolus Doppler Wind Lidar satellite mission are expected to improve the understanding of atmospheric processes
and the quality of weather forecasts (Baars et al, 2020) through data assimilation (Horányi et al., 2015). The sharp land-sea transitions and the scattered islands might offer an advantage in acquiring ground based observations, as in the case of Qendresa passing over Malta, but also a relatively wide radar coverage (e.g. McTaggart-Cowan et al., 2010). On the other hand, observation campaigns are rare and seldom dedicated to cyclones. Consequently, it is a challenge to validate the capabilities of simulations, or even to validate important cyclone characteristics such as central pressure, profiles of the
atmospheric state and frontal structures. Two other important issues regarding observations are accessibility and lack of spatial homogeneity. For instance, observations over North Africa are scarce and not easily accessible due to the lack of participation in joint international projects. This poses a challenge for understanding the uptaking of dust by cyclones and its interactions with cyclone dynamics. Inclusiveness and international coordination should be prioritized in future projects. A final issue concerns the direct transposition of diagnostics and concepts from the fields of tropical cyclones and mid-latitude





storms to Mediterranean cyclones. For instance, Hart diagrams need to be adapted to the latitude and size of Mediterranean cyclones (Picornell et al., 2015), but also to the finer horizontal resolutions used by state-of-the-art models and reanalysis. In addition, the direct applicability of WISHE and tropical transition theories to medicanes should be revised according to recent results. New research efforts should address these issues taking advantage of the increasing availability of gridded observations and new reanalysis at fine grid spacings. This renders the systematic analysis of cyclone dynamics and

processes more accessible than a decade ago and opens new opportunities for the analysis of cyclone dynamics, data assimilation and models validation.

**6.6 Reducing error and uncertainty in numerical weather prediction**

As numerical model resolution increases and nonlinear cyclogenetic processes become explicitly resolved, new strategies to short and long term observation targeting for Mediterranean cyclones are needed. Indeed, although baroclinic processes have

a substantial linear component, the aforementioned determinant role of nonlinear diabatic processes in the precise evolution of mature Mediterranean cyclones sets new challenges. New sensitivity calculation methods must account for nonlinear links between predictions and predictors (Jourdan and Loubière, 2020) and most likely exploit the advances in artificial intelligence to uproot complex sensitivity links from high resolution ensembles. Furthermore, most case studies up to now have focused on medicanes only and are based on the ECMWF ensemble forecasting system solely. This calls for a

comparative study of the predictability of different types of Mediterranean cyclones and in different forecasting systems. In fact, case studies based on complex modeling strategies are still necessary to better understand processes but do not necessarily help to improve forecasts, which remain insufficiently constrained. Relevant mesoscale observations may be lacking to prescribe initial conditions which are often downscaled from global models in the absence of appropriate data assimilation systems. Therefore, uncertainty might be high in the large scale conditions, jeopardising both accuracy and

reliability of cyclogenesis. This calls for more research to identify sources of uncertainty for Mediterranean cyclones. In these regards, stochastic methods should focus more on the specific processes that hold high uncertainty in cyclone dynamics, but also on the exchanged variables at the interface of coupled models. Finally, new research is also needed to identify useful observation operators and covariance relationships that help extract more accurate information from new observational means (mostly remote sensing) and produce better descriptions of maritime environments of Mediterranean

cyclones.

**6.7. Extending the scope of climatological studies for Mediterranean cyclones**

The analysis of Mediterranean cyclone tracks in a climatological context has been one of the most investigated topics. Despite the different approaches, reanalysis datasets and cyclone tracking methods, recent studies present high agreement on the tracks density, seasonal cycle and favourite locations of cyclogenesis. Several subtypes of cyclones have also been

identified  and a high number of studies have addressed their climatological aspects. Nevertheless, open questions still remain on the future evolution of Mediterranean cyclogenesis and associated  impacts. In these regards, the special case of



medicanes has drawn most of the attention using a plethora of different criteria for their detection. However, the rarity of medicanes and the lack of agreement in their physical definition prohibits a solid interpretation of the results. Furthermore, such high focus on medicanes has eclipsed the importance of analysing the future trend of other intense Mediterranean

cyclones. New studies need to focus on the multiscale processes that affect the development of Mediterranean cyclones. This notably includes the future evolution of the large scale conditions leading to RWB and the resulting southern deviations of the polar jet. Although this is the main mechanism triggering Mediterranean cyclogenesis, its systematic analysis is still missing from the state of the art. Finally, new studies should also focus on the contribution of cyclones to the variability and extremes of the regional climate. This includes the establishment of cyclones' relationship to the regional water cycle and

risks related to heavy precipitation events, dust transport and storm surges.

### 6.8 Assessing the various impacts of Mediterranean cyclones

Impacts are inevitable if an intense cyclone occurs in areas of high vulnerability. The recent medicane Ianos is a characteristic example, affecting Greece on 18-19 September 2020 with 4 fatalities and devastating impacts due to heavy rainfall, gusty winds, and high waves (Lekkas et al, 2020; Lagouvardos et al. 2021). However, the relationship between

cyclone intensity and the severity of impacts is not linear. Various factors and characteristics should be considered and therefore an improved knowledge about the intrinsic relationship between Mediterranean cyclones and high-impact weather is needed. In this context, the state-of-the-art suffers from the lack of systematic quantification of cyclones contribution to Mediterranean high impact weather. In fact, there are many studies focusing on heavy precipitation events and cyclones; conversely, other relevant risks associated with  cyclones, such as windstorms and sea state, have gained less attention.

Especially concerning the sea state, few studies on air-sea interactions focused on impacts. For instance, the wave field associated with cyclones is usually a mixture of wind and swell due to the spatial inhomogeneity of intense surface winds and their directional variability among the different cyclone sectors (Ponce de León and Bettencourt, 2019). Extreme storm surges may result from the superposition of different factors directly related to cyclones, such as pressure drop, winds, meteotsunamis, waves, superimposed to the astronomical tide (Lionello et al., 2020; Ferrarin et al., 2021). In addition,

intense rainfall interacts with the waves attenuating the development of shorter waves at the high frequency tail (Cavaleri and Bertotti, 2017). The complexity of in-cyclone wave patterns is still an open question that calls for extensive research to increase predictability and assess the risk of high waves in maritime transportation and coastal structures. Considering the high impacts that Mediterranean cyclones produce in terms of human losses, property and infrastructure damages, the need to step from hazard forecasts (such as heavy rainfall, wind gusts, etc.) to impact-based forecasts is essential (Taylor et al,

2018, Zhang et al., 2019). In this context, the World Meteorological Organisation (WMO) World Weather Research Program (WWRP) has launched in 2015 the High-Impact Weather Project (HIWeather, http://hiweather.net) that aims at enhancing the communication of the high impact weather forecasts to increase resilience of societies. Societies resilient against high impact weather need impact-based warning systems, structural protection, and proper training and education. This is

imperative to reduce fatalities and facilitate recovery from the consequences of high impact weather events, including those
related to Mediterranean cyclones.

**7 Conclusion**

This article performed a review of the status of knowledge in the broader field of Mediterranean cyclones and outlined future research directions. The coordination of community efforts deems necessary to advance the field of Mediterranean cyclones as a whole and, in most cases, these efforts need to include interdisciplinary approaches. In these regards, existing
knowledge, tools and methods are needed to balance the asymmetric progress in different subdomains of the field and in different regions within the Mediterranean. For instance, cyclones prediction in weather and climate scales are both strongly dependent on models. However, forecasting cyclones is still lacking the substantial amount of multi-model approaches that have been performed in climatological studies. In parallel, new methods need to be developed for newly identified challenges in the field, or rather under addressed research topics need to be addressed. Among others, the understanding of
dust or sea waves interaction with cyclone dynamics and their consequent impacts demand dedicated observations and model adaptations, both performed in an interdisciplinary framework. Therefore, the main contribution of this article is to highlight the research topics that deepen our knowledge on Mediterranean cyclones and to provide useful guidelines to the international community efforts, especially those that have been coordinated in the framework of the COST Action CA19109 Medcyclones (European network for Mediterranean cyclones in weather and climate), recently started.

**Code and data availability**

No new analysis has been performed in this article. All presented results are already published in previous papers.

**Author contribution**

All authors contributed to writing, commenting and reviewing the manuscript. Emmanouil Flaounas, Silvio Davolio, Shira Raveh-Rubin, Florian Pantillon and Mario Marcello Miglietta conceptualized the structure and organized the content of the
aticle.

**Competing interests**

The authors declare that they have no conflict of interest.



## Acknowledgements


This article is based upon work from COST Action CA19109 "MedCyclones", supported by COST - European Cooperation in Science and Technology (www.cost.eu)

S. Davolio and M. M. Miglietta were supported by the project "Climate change: risk mitigation for sustainable

development" funded by the Italian Ministry for Education, University and Research (MIUR)

M. A. Gaertner has been supported by the Spanish Ministry of Science, Innovation and Universities, the Spanish State Research Agency and the European Regional Development Fund, through grant CGL2017-89583-R (IBERTROPIC project).


S. Khodayar is supported by the Talented Researchers Support Programme - Pla GenT - CIDEGENT by the Ministry of Innovation, Universities, Science and Digital Society, Generalitat Valenciana (GVA), project MED-EXTREME (CIDEGENT/2018/017).

M.Reale has been supported in this work by OGS and CINECA under HPC-TRES award number 2015-07 and by the project FAIRSEA (Fisheries in the Adriatic Region - a Shared Ecosystem. Approach) funded by the 2014 - 2020 Interreg V-A Italy - Croatia CBC Programme (Standard project ID 10046951).

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

**Figures**
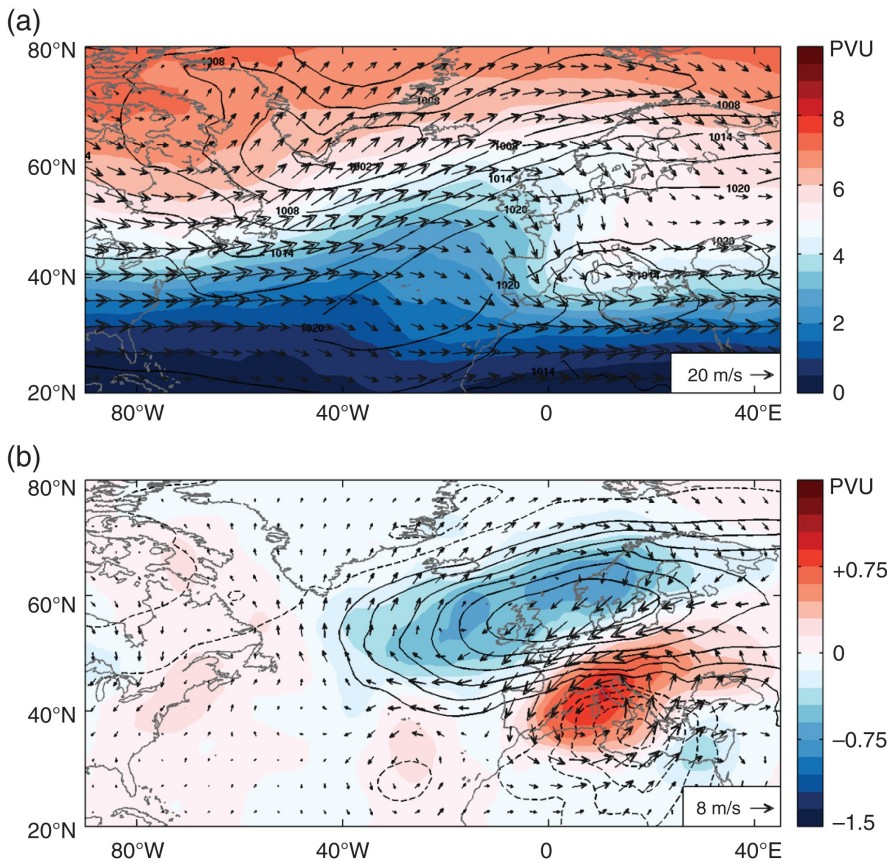

**Figure 1: (a) ERA-Interim average fields of sea-level pressure (black contours), PV (shaded), and wind (arrows) on the 330-K isentropic surface, centered at the time of maximum intensity of the 200 Mediterranean cyclones. (b) as (a) but for monthly anomalies (contours every 1 hPa, dashed for negative and solid for positive values). Reprinted from Raveh-Rubin and Flaounas (2017).**





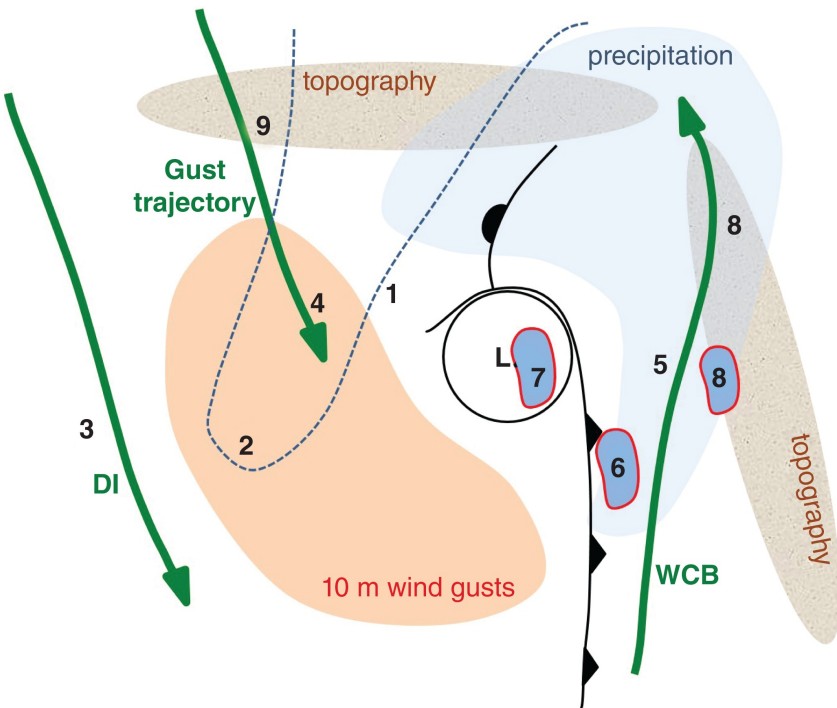

**Figure 2: Schematic illustration of possible features related to combined precipitation and wind impact of a cyclone in the Mediterranean, based on the five events studied in Raveh-Rubin and Wernli (2016). Note that this schematic does not represent any individual event, but rather summarizes the variable possible features of importance. The cyclone centre is denoted by the letter 'L', accompanied by a cold and a warm front. Shading shows areas with precipitation impact (light blue), and with 10 m gust impact (light red). Areas with convection (and thus co-located precipitation and wind gust impact) are shaded blue and encircled by a red line. High topography is represented by grey regions. The 320 K 2-PVU contour is shown as a dashed line and typical WCB, DI and gust trajectories are denoted as green arrows. The numbers mark the location of: (1) a prominent upper-level feature (PV streamer/trough/ridge), (2) tropopause fold and downward momentum transfer, (3) DI trajectories around high gusts, (4) low-level gust trajectory, (5) WCB slantwise ascent associated with precipitation, (6) convective precipitation at cold front, (7) convective precipitation in cyclone centre, (8) orography enhancing precipitation (and/or convection), (9) orography accelerating gust trajectories. Reprinted from Raveh-Rubin and Wernli (2016) with permission.**




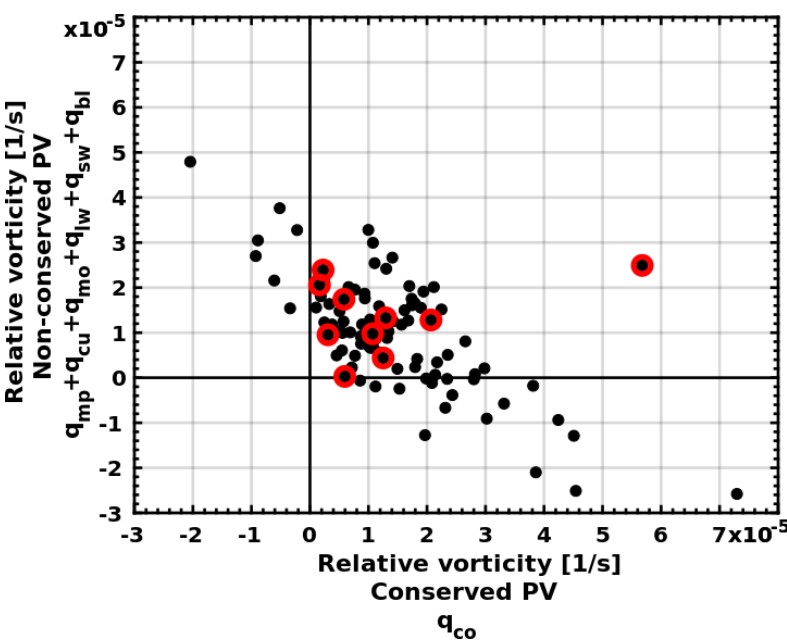

**Figure 3: Scatterplot relating the contribution of different PV sources to the relative vorticity at 850 hPa, in the centre of 100 intense cyclones: Contributions of conserved, adiabatically transported PV is shown in the x-axis and non-conserved, diabatically-produced PV is shown in the y-axis. *Red circles* depict ten medicane cases. Reprinted from Flaounas et al. (2021).**

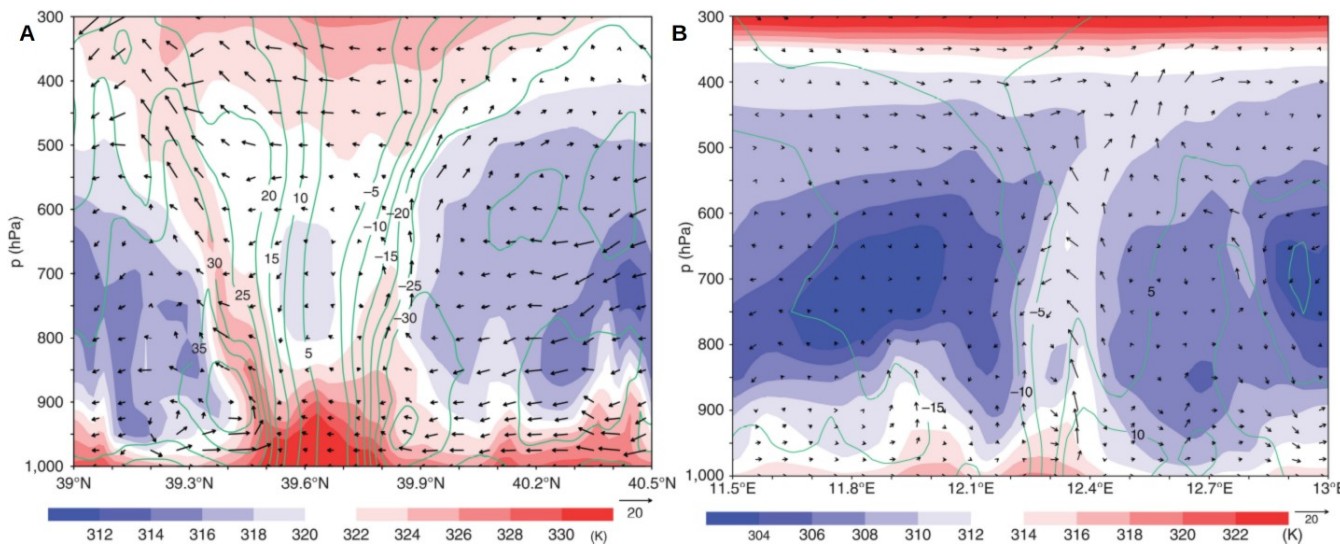

**Figure 4: (a) The October 1996 case: Vertical cross-section of $\theta$e (colours), storm-relative winds (vectors), absolute momentum (lines, contour interval=5m/s; zero not shown) near the cyclone centre (longitude=12.45∘E) (WRF-model run, 3 km grid spacing), at 1000 UTC 9 October 1996; (b) as in (a), but for the December 2005 case (latitude = 34.1°N; 0600 UTC 15 December 2005.. The wind vectors are drawn every five grid points in the horizontal. Typical TC features can be identified, such as upward motion along isosurfaces of absolute momentum, large-scale moisture convergence in low levels, and near moist-neutrality for ascending**
**particles. Reprinted from Miglietta and Rotunno (2019) with permission.**



**Figure 5: ECMWF forecasts at 1200 UTC on 26 September 2006: PV at 300 hPa (shaded, PVU = 10 −6 K m −2 s −1 kg −1 ), wind at 300 hPa (vectors, for winds larger than 30 m s −1), and mean sea-level pressure (solid lines, every 4 hPa between 976 and 1008 hPa). Forecasts are at (a) t + 12, (b) t + 24, (c) t + 36, (d) t + 60, (e) t + 84, and (f) t + 108. The figures (when applicable) indicate the local MSLP minima over the North Atlantic Ocean and the Mediterranean Sea associated with the ex-tropical cyclone Helene and the medicane, respectively. Reprinted from Chaboureau et al. (2012) with permission.**







**Figure 6: Control surface divergence, horizontal winds and mean sea level pressure (a-c) and hourly accumulated precipitation together with mean sea level pressure (d-f) depicted at a-d) 00, b-e) 06 and c-f) 08 UTC on 31 October 2012. Red circle highlights locations of low-level convergence associated with triggering of the secondary cyclone. Reprinted from Carrio et al. (2020).**



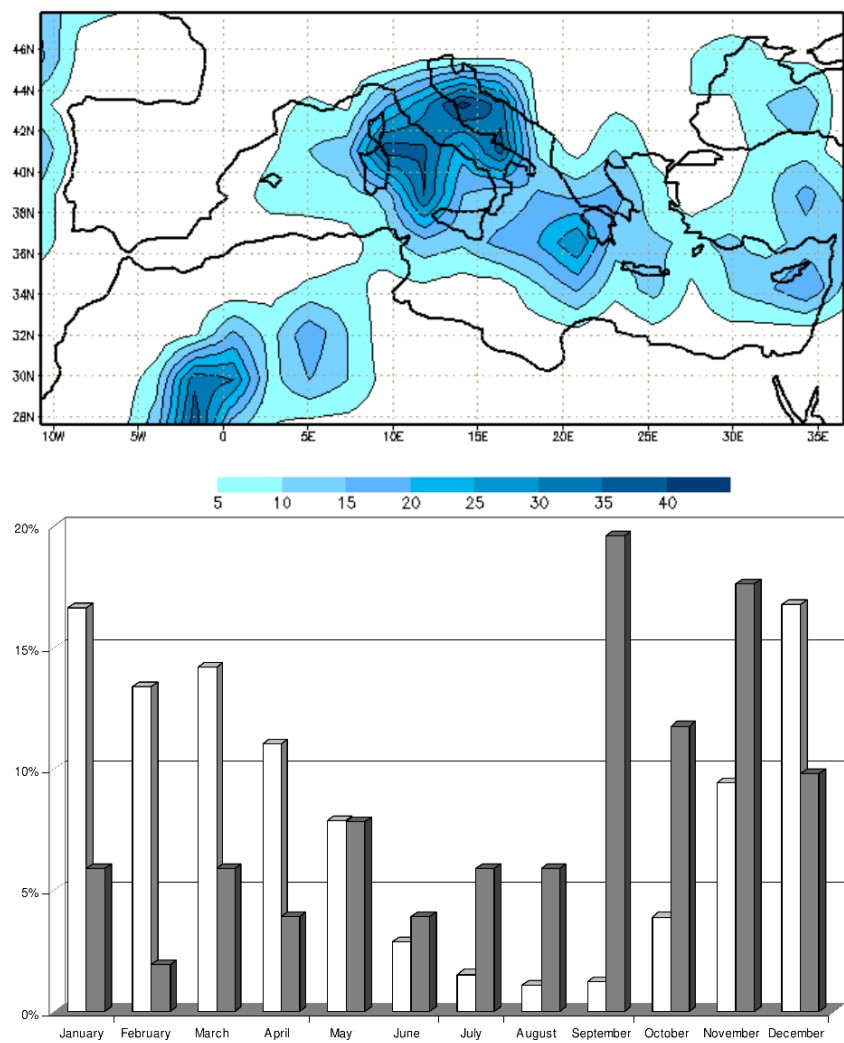

**Figure 7: (Upper panel) Number of intense cyclones over the 45 years ERA-40 period at mature stage per squares of 2.25o×2.25o. (Lower panel) Monthly frequency of intense cyclones in the ERA-40 analysis 1957–2002 (white) and monthly distribution of episodes in the MEDEX "list of selected cases" (grey). Reprinted from Homar et al. (2007).**





**Figure 8: 24-hours accumulated precipitation (grey shading) provided by the Tropical Rainfall Measuring Mission (TRMM), areas of lightning activity (red contours, denoting areas with at least five impacts) from ZEUS (National Observatory of Athens), and areas of deep convection (green contours) as diagnosed from the Microwave Humidity Sounder (MHS) radiometer. Reprinted from Flaounas et al. (2016) with permission.**







**Figure 9: (top row) Integrated water vapor (mm; color shading) and geopotential height at 500 hPa (m; contours), (bottom) integrated vapor transport (kg m-1 s-1; color shading and arrows) and mean sea level pressure (hPa; contours) at 1200 UTC (left) 28 and (right) 29 Oct 2018, as simulated by BOLAM NWP model (© American Meteorological Society. Used with permission).**



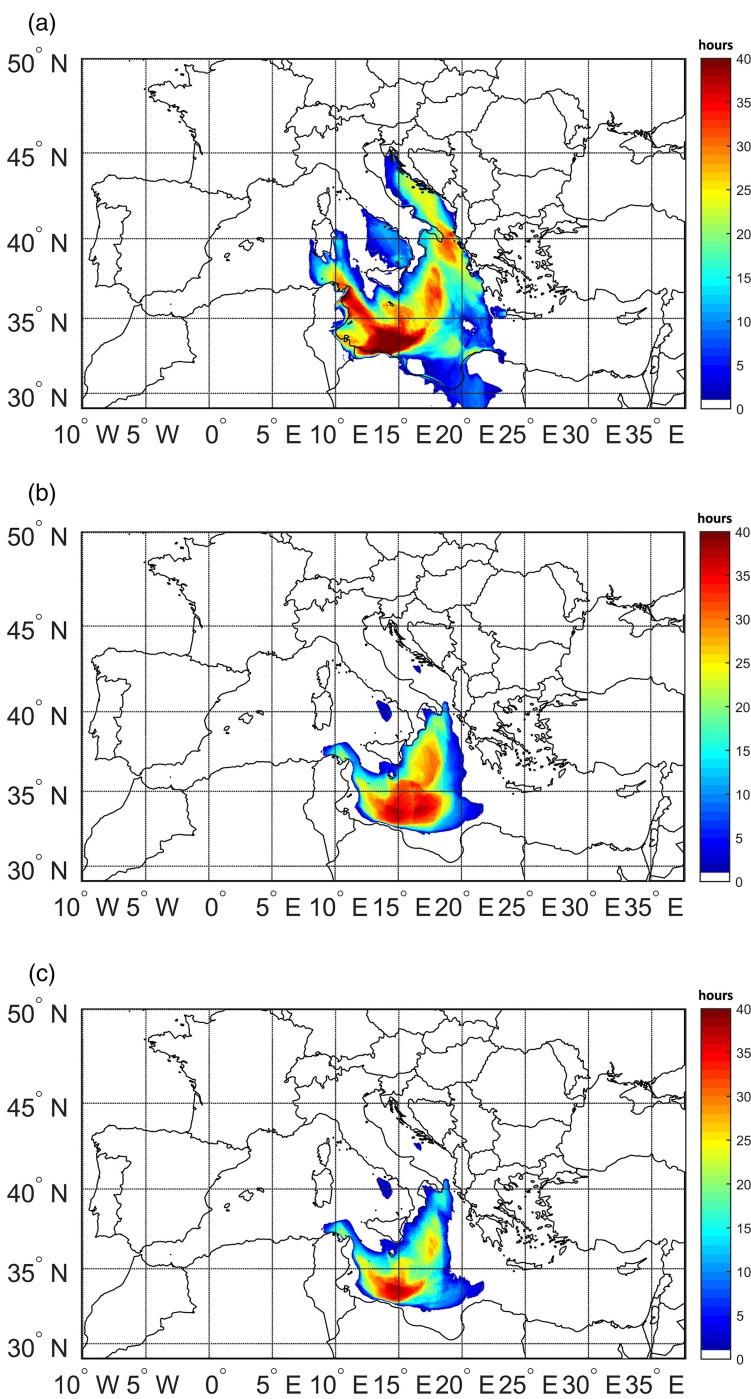

**Fig 10 (a) Duration in hours per grid point where 925 hPa wind speed exceeds the threshold of 17.5 m/s during the lifetime of medicane Qendresa (2014). (b) as in (a) but for significant wave height exceeding 4 meters. (c) as in (a) but for the combined threshold values of both wind speeds and waves height. Fields derive from coupled waves-atmopshere model. Reprinted from Patlakas et al. (2020) with permission.**






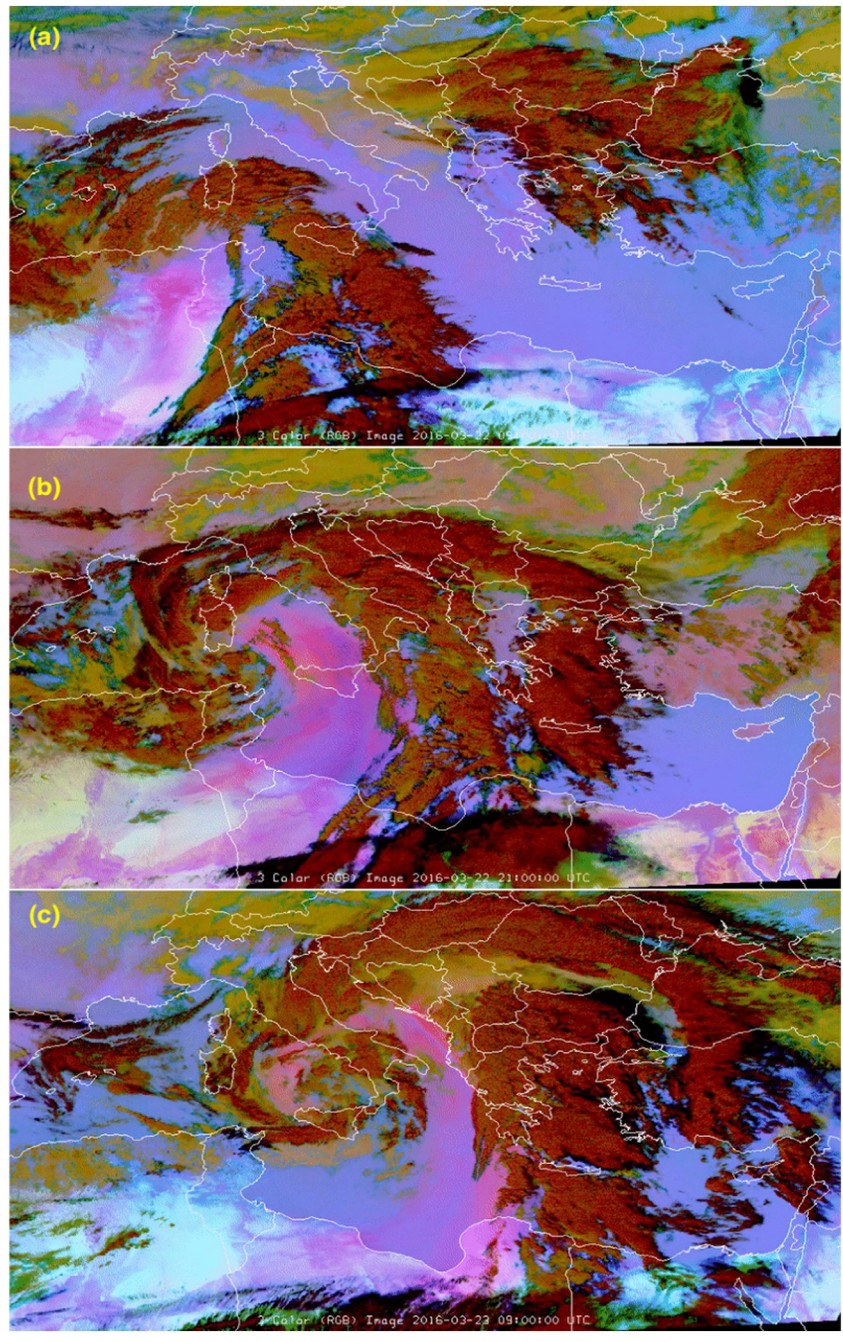

**Fig. 11 MSG-SEVIRI dust product, showing dust (pink/purple), clouds (brown/orange), and differences in surface emissivity retrieved in absence of dust or clouds (light blue/blue) on (a) March 22 at 09:00 UTC (b) March 22, 21:00 UTC (a), and (c) March 22, 09:00 UTC of 2016. Rizza et al. (2018a) with permission.**
