# Peer review of "Mediterranean cyclones: Current knowledge and open questions on dynamics, prediction, climatology and impacts"

_Weather and Climate Dynamics, 2021_

## Author Response (AR1)

**Reply to the Reviewers' comments**

**Reply to first and second Reviewer's comments**

*1) This paper reviews the current knowledge on Mediterranean cyclones and the open future research questions. I found that the paper provides a complete and in depth description of the characteristics, genesis, tracks, dynamics of Mediterranean cyclones, including various subcategories, on a synoptic and climatological basis. It includes the whole range of previous studies without any exception, focusing on new findings, consistencies /inconsistencies, and queries. I think that the paper is excellent and provides a fundamental and robust background for many researchers working on Mediterranean weather and climate. I suggest that it can be accepted in its present form. The only comment I have refers to the abstract. I think that it should be reorganised to reveal main findings and challenges and should be so generalised.*

*2) This paper compiles numerous previous works in which different aspects of Mediterranean cyclones have been studied. Mechanisms of different scales that play an important role in the formation and evolution of cyclones are analysed. Efforts made to improve forecast are reported and numerous climatological studies of cyclones in the Mediterranean are referenced, as well as the high impact weather that is frequently associated. Special attention is devoted to medicanes, small cyclones with tropical characteristics that are the subject of numerous recent studies. Some little-known aspects are highlighted in the text and are presented as the object of possible future work. For this reason, I highly recommend the publication of this paper, which can be a good guide to advance knowledge of Mediterranean cyclones and which poses challenges for future studies.*

We would like to thank both Reviewers for carefully reading the manuscript and for providing such a positive feedback. In response to the first Reviewer, we have revised the abstract. We now make more explicit reference of the sections content.

**Reply to third Reviewer's comments**

*This review is a thorough overview of Mediterranean cyclones and the many aspects of their climatology, dynamics, forecasting, and impacts. It gives a clear summary of the open research questions and proposes areas for future research focus. The paper is well-written and enjoyable to read, and contains a wealth of references and important information. I have a few minor comments and suggestions.*

We thank the Reviewer for the positive feedback and for the helpful comments to improve the manuscript. Please find below our replies to each comment

1. *I suggest to move the climatology section to before the dynamics section. This is because as I was reading the dynamics section, I found myself wondering about when the features most commonly occur and where. This information is provided later in the climatology section, but I think would give a better flow being first.*

Acknowledging that this is an unconventional way to organise the manuscript, in the revised version we have rearranged the sections. However, both sections on "climatology" and "dynamics" have been shortened in the revised version by reallocating the subsections on "cyclone subtypes" and "medicanes" within a new section. This new section comes after the dynamics to resolve the issue of explaining medicane and subtypes without having first outlined all necessary dynamical processes. Finally, the manuscript continues with the sections on forecast and impacts. In the end of this document, we present the revised table of contents.

*2. In order to summarise the many studies looking at future changes in Mediterranean cyclones, I suggest including a table or a schematic. Section 4.4 gives a lot of information about the different studies that have investigated this question, and it is not easy to discern the overall picture.*

We thank the Reviewer for the suggestion. All cited studies use different datasets, time periods and methods. We agree that a table would ease access to extensive information but would also provide numerous details that risk to eclipse the main result of the future decrease of cyclone occurrences. We have substantially revised section 4.4 to simplify the text and clarify the messages.

*3. For WCD, either British English or American English are accepted. Most of the paper is written in British English, but there are a few inconsistencies. Please check through for consistency of spelling of e.g. modelling/modeling, characterise/characterize, centre/center, favouring/favoring, Also consistent spelling of parameterization.*

Spelling has been consistently changed to British English.

*Some typographical and text suggestions:*
*1. Line 76: "to" -> "of".*

Done

*2. Line 129 and 131: The abbreviation DI has already been defined so "dry intrusion" can be shortened. Also on line 131 – it states that frontal precipitation is weaker on average, but I'm not clear in comparison with what?*

Done

*3. Line 132: Could the weak precipitation associated with the suppression of convection by the DI?*

Done

*4. Line 209: I'm not sure you need a separate subsection here – this is still about the special case of medicanes.*

Title has been revised to:

*"The special case of medicanes: processes and classification"*

*5. Line 216: In the Miglietta and Rotunno paper, how many storms were put into each group? In other words how robust is the classification?*

The classifications in Miglietta and Rotunno (2019) is based on the analysis of three cyclones, which however have characteristics representative of the different case studies present in the literature. We changed the final sentence of the first paragraph that introduced the three groups to:

*"Based on the limited number of cyclone cases identified as medicanes, Miglietta and Rotunno (2019) recently categorised medicanes in three main representative groups with similar dynamical*

*processes during their mature stage (a similar classification was provided recently in Dafis et al., 2020):"*

6. *Sections 2.5 and 2.6 – this is the place in the paper that I particularly was wondering about the climatology and when medicanes preferentially happened.*

The manuscript structure has been reorganised. Please refer to our reply to the first major comment.

7. *Line 362-365: This is a long sentence – consider revising.*

Thank you for this comment, this part has been simplified.

8. *Line 398: "secondary cyclone" is not clearly defined in the paper. In previous sections is seems to be associated with lee cyclogenesis. It would be good to be consistent with terminology.*

"Secondary" has been removed and the whole paragraph has been revised as follows:

*"As an example of the usefulness of high resolution simulations, Carrió et al. (2020) analysed the genesis of a rather small cyclone with a characteristic scale of few kilometres that was embedded within the broader cyclonic circulation of a Mediterranean cyclone. The latter travelled along the northern half of the Western Mediterranean and produced heavy rainfall over Italy (Flaounas et al., 2016). Figure 6 shows the development of the small-scale cyclone and its clear relationship to convection. Carrió et al. (2020) used piecewise PV inversion to further show that diabatic heating within the small-scale cyclone was crucial to intensify the encompassing Mediterranean cyclone. The high resolution outputs of AROME-WMED were the key to diagnose the intense winds produced by the embedded cyclone that affected the Balearic Islands."*

9. *Line 512: "Hart diagrams" have not really been defined in the paper.*

Hart diagrams were changed to "cyclone phase-space analysis (Hart, 2003)"

10. *Line 524: Sentence beginning "Their formation…" -> "Their formation over the arid areas of North Africa **means** Sharav cyclones **have** a low moisture content and **are** associated with dust storms."*

Thank you for this comment. The phrase has been corrected as suggested.

11. *Line 593: Suggest to start the sentence with "Changes in cyclone frequency/intensity…"*

Thank you for this comment, the phrase has been changed as suggested.

12. *Line 851: I think "vital" or "important" rather than "mandatory".*

Thank you for the suggestion. "Mandatory" is changed to "vital".

**Reply to Editor's comments**

*This paper provides a thorough review of Mediterranean cyclones, covering many different aspects of the topic and ending with a very useful summary of open research questions. The authors are to be congratulated for synthesising a large literature into a coherent and readable review paper. Consistent with the views of the anonymous referees, I believe the paper should be acceptable for publication in WCD after some minor revisions.*

Thank you for your support during the peer review process, for the careful reading of our paper and for providing fruitful comments.

*Having been through the paper myself, I am inclined to agree with referee 3 that the climatology section would be useful to read before the dynamics section, so I invite the authors to consider this.*

The manuscript structure has been reorganised. Please refer to our reply to the first major comment of the third Reviewer.

*In addition, I have some further comments for the authors to consider:*

*1. The need for interdisciplinary work is highlighted in several places, including the abstract, but I wonder if the authors could be more specific regarding which disciplines they are thinking of. Does this refer to the weather and climate communities, or others?*

We have revised the conclusion to be more specific:

*"The coordination of community efforts deems necessary to advance the field of Mediterranean cyclones as a whole and, in most cases, these efforts need to include interdisciplinary approaches. Such approaches would demand collaboration between climate and weather communities, but also between researchers working on oceanography, atmospheric composition and dynamics. The combination of existing knowledge, tools and methods could balance the asymmetric progress in different subdomains of the field and in different regions within the Mediterranean. For instance, ...."*

*2. Cyclogenesis is deemed 'exceptionally frequent' and 'one of the highest in the globe', with several references to Petterssen. This feels a little strong, and potentially not a consensus view across studies. For example, the Hoskins and Hodges (2002) climatology shows several regions of more frequent cyclogenesis around the hemisphere.*

Thank you for this comment.

Hoskins and Hodges (2002) confirmed former results by Petterssen who marked the Mediterranean as a region of highest incidence of low pressure centers, especially in winter. Given the smaller synoptic scales involved in this area (and thus relatively small scale of the storms), the Mediterranean maximum is more marked in vorticity. Indeed, the identification of regions with frequent cyclogenesis may vary according to the applied method. For instance, Fig. 1 of Neu et al., (2013) shows different amplitudes of track densities in the Northern Hemisphere. Nevertheless, the majority of methods agree that the Mediterranean stands out as a distinct area of high tracks density.

In fact, the density of cyclogenesis is the peculiar characteristic of the Mediterranean, (and not the cyclone frequency or the track density which is extremely higher in the Pacific). It means high frequency of cyclogenesis concentrated in a relatively small areas, especially on the lee side of the

mountain chains. The cyclogenesis density clearly emerges in the Hoskins and Hodges analysis, together with other few areas around the globe.

In any case, we agree that the statement we use in the manuscript is rather strong and therefore we have revised this part to :

*"The Mediterranean basin is a relatively small region, but of unique and complex geography. It is characterised by a nearly enclosed basin with sharp land-sea transitions. It is surrounded by high mountain chains and it has been long ago identified as a distinct region in the globe of frequent cyclogenesis (Petterssen, 1956)."*

Neu et al.: IMILAST: A Community Effort to Intercompare Extratropical Cyclone Detection and Tracking Algorithms, 94, 529–547, https://doi.org/10.1175/BAMS-D-11-00154.1, 2013.

*3. The role of baroclinic instability is noted in section 2.1, but can more be said about the potential for this. Eg how strong is the lower tropospheric baroclinicity compared to the main storm tracks? Why is this not expected to significantly affect the surface cyclone (line 105)?*

Low level baroclinicity in the Mediterranean is rather weak compared to the one in the storm tracks and therefore any baroclinically induced temperature anomalies are naturally expected to have a comparably weaker forcing to cyclones development in the Mediterranean. In addition, several case studies and numerical sensitivity tests in the recent past (thoroughly documented in this review paper) show the dominating impact of diabatic processes and large scale forcing on cyclones development. Therefore, we believe that there is little doubt about the weak role of surface baroclinicity in cyclones development. Nevertheless, we agree that the issue of surface baroclinicity is not thoroughly addressed in the field of Mediterranean cyclones (along with the analysis of frontogenesis). We are now more explicit on the study of Flocas et al. (2000) and highlight the need for additional research:

*"Compared to cyclones over open oceans, Mediterranean cyclogenesis takes place across lower-tropospheric temperature gradients where baroclinicity is partly due to land-sea surface temperature contrast (Lionello et al., 2006; Trigo et al., 2002). Therefore, the low-level atmospheric baroclinic forcing to the surface cyclone is expected to be comparably weak. Indeed, Flocas (2000) used PV inversion diagnostics to show that surface temperature anomalies contributed the least to the surface circulation of an intense Mediterranean cyclone over the Aegean Sea with respect to mid tropospheric and upper tropospheric PV anomalies. The latter PV anomalies were shown to dominate the development of the cyclone, however more studies are needed to thoroughly address this issue."*

*4. The negative relationship in Figure 3 is very interesting - can more be said about the reasons for this? Presumably it is not just an artefact of the method of cyclone selection?*

Thank you for this comment. Figure 3 shows very recent results and therefore the questions it raises can only be addressed in future studies. In their study, Flaounas et al. (2021; where the figure is taken from) used a climatology of 100 cyclones. In their section 3 previous climatological studies were compared to these 100 cyclones and it was shown that this cyclones sample was indeed representative of intense Mediterranean cyclones in terms of time of occurrence, intensities and dynamics. Therefore it is less likely for the negative trend in Fig. 3 to be an artefact of the method of

cyclone selection. We added the following to provide a possible perspective research that could shed some light in the issue of the negative trend:

*"Such a negative relationship between baroclinic and diabatic forcing is still an open question and needs to be addressed with additional studies. For instance, numerical sensitivity tests should investigate the possible role of convection in producing negative PV anomalies in the upper troposphere and therefore reducing the contribution of upper tropospheric PV streamers to the development of surface cyclones."*

*5. Section 2.4: Have any studies quantified the fraction of cyclones which depend on the orography? In figure 7 there seem to be many cyclones located away from orography.*

Figure 7 by Homar et al (2006) considers only the most intense cyclones and refers to their mature stage. It does not refer to the genesis stage that usually take place closer to the mountain chains, as can be inferred by Mediterranean cyclone tracks.

At the genesis stage, the most intense cyclones would occur even without the presence of orography, due to the strong baroclinic forcing. However, the evolution of the cyclone would be different (displaced cyclone center, different growth rate etc.). In fact, it is rather difficult to disentangle orographic contribution to cyclones development from other atmospheric processes (e.g. convection) and therefore to quantify the fraction of cyclones whose formation depends "exclusively" on orography. The presence of the orography is a bottom boundary condition: it may be enough to produce relatively shallow cyclones alone; very often it contributes to the evolution of the cyclone (cyclogenesis location, trajectory, growth, etc.). We included the following as a conclusion to Section 2.4:

*"Thus, orography constantly affects cyclonic systems in the region, depending on the development stage and location of cyclone centers. Direct effects include lee-cyclogenesis and deepening, close to mountain chains, while indirect effects include contribution to the development of the cyclone through the interaction of the atmospheric flow and cyclones themselves."*

*6. The discussions on medicanes vs other cyclones are interesting. Could the seasonality be used more here, given the comparison to tropical cyclones which have quite distinct seasonality compared to extratropical cyclones?*

Thank you for this comment. The physical definition of medicanes is still an open question and the number of cases qualified as medicanes is still rather small to compare with tropical cyclones in a climatological context. Nevertheless, we agree that seasonality could be a potential point of difference between and thus we added the following in lines 240-244:

*"Finally, seasonality: cases qualified as medicanes present a distinct climatology where their peak of occurrence is observed between late autumn and winter, i.e. slightly anticipated compared to extra-tropical cyclones. This lag time is typically attributed to the warm SSTs that favour stronger convection in cyclones centre than in winter."*

*7. Figure 5 is just mentioned briefly. Could more be said about what the reader should take from it?*

We changed the corresponding part of the text to :

*"Several case studies focusing on medicanes have shown that large scale conditions often pose a challenge to the predictability of Mediterranean cyclogenesis. For instance, Chaboureau et al. (2012) suggested a link between the poor skill of operational forecasts of the European Centre for Medium-Range Weather Forecasts (ECMWF) for the September 2006 medicane and the extratropical transition of Hurricane Helene taking place upstream. To gain deeper insights into the large-scale atmospheric circulation relevant to the poor forecast of the medicane, Fig. 5 compares different representations of Hurricane Helene and an event of RWB that eventually triggers cyclogenesis downstream in the Mediterranean. While short-term forecasts well predict both Hurricane Helene and the medicane until 36 h (panels a-c), forecasts at longer lead time progressively miss the position of the hurricane over the Atlantic, the amplification of the Rossby wave train, and the occurrence of Mediterranean cyclogenesis (panels d-e). The link between these dynamical features was confirmed by Pantillon et al. (2013) using ECMWF ensemble forecasts. The fraction of ensemble members capturing the medicane formation dropped for lead times beyond 2 days. This was due to the sensitive phasing between the hurricane and the Rossby wave train, a situation characterized by high forecast uncertainty as measured by the ensemble spread (Anwender et al. 2008)."*

8. The predictability sections are very useful but I wonder if a clearer distinction could be drawn between the intrinsic predictability of the cyclones and the realised predictive skill of the forecast systems (Eg pages 8-10). If an event is not captured by many ensemble members beyond a certain lead time, is that because of model error or just because it is an uncertain situation in reality?

Thank you for this comment.

It is quite difficult to make a clear distinction between "intrinsic" and "practical" predictive skill in numerical weather prediction. Currently there is no objective definition of intrinsic predictive skill and previous attempts are based on current paradigms (Zhang et al. 2019). We are not aware of a relevant study for Mediterranean cyclones and we agree that this is an important aspect that needs to be addressed in the future. We changed the introductory paragraph of Section 3 to:

*"The close link between Mediterranean cyclones and severe weather phenomena has historically been at the basis of international coordinated efforts devoted to improve the practical predictability of the broad range of these cyclones. Despite the intrinsic predictability limits of these systems, linked to the predictability of midlatitude weather (Zhang et al. 2019), the practical forecast limits of socially relevant aspects of Mediterranean cyclones, such as associated winds and precipitation, is assumed much shorter. This can be attributed to:…"*

*Zhang, F., Sun, Y. Q., Magnusson, L., Buizza, R., Lin, S., Chen, J., & Emanuel, K. (2019). What Is the Predictability Limit of Midlatitude Weather?, Journal of the Atmospheric Sciences, 76(4), 1077-1091.*

9. It might be nice for the climatology section to connect briefly with the storm track perspective of the general circulation. For example, does the Mediterranean storm track emerge from filtered variance analyses such as of eddy kinetic energy, and is it thought to play any role in the global circulation? Eg do Mediterranean cyclones achieve much poleward heat transport?

Thank you for this comment. We added the following to section 4.3 (now 2.2):

*"From an energetic perspective, Nasr-Esfahany et al. (2011) and Rezaeian et al. (2016) selected critical positive NAO and negative NAO winter months spanning 1950–2011. They showed that*

*there is energy transfer from west Atlantic and North Europe to the Mediterranean region, which is stronger during the negative phase of NAO. Conversely, during the positive phase, more energy is radiated from the central Mediterranean towards the Red Sea and North Africa. This indicates that while in the negative phase, the Mediterranean is more affected by the North Atlantic storm track, it forms an independent centre of action in the positive phase radiating energy southeastwards. Wave activity is expected to spread to the Mediterranean storm track through equatorward refraction process from the North Atlantic wave packets toward the North African and South Asian regions (Hakim 2003). In accord to that, Ahmadi-Givi et al. (2014) studying the propagation of Rossby wave packets linking the North Atlantic and Mediterranean storm tracks in winter 2004/05, demonstrated the presence of energy flux on the upstream and downstream sides of the troughs associated with the wave packets. This suggests that downstream wave packet over the Mediterranean develops by receiving energy from the upstream wave packet in the North Atlantic which at the same time weakens."*

*10. Section 4.3 is rightly focused on winter, but is there also any literature on links between cyclones and the summer NAO?*

When considering intense cyclones, summer time (JJA) cyclogenesis is considerably reduced in the Mediterranean. This is due to the Atlantic storm tracks migrating towards the north, making thus the intrusion of troughs or PV streamers in the Mediterranean less likely to occur. As a result, intense summer cyclones are rare events and naturally NAO studies have mainly focused on the winter period. We added the following to highlight the need for additional studies that extend the analysis to summer:

*"Storm track and NAO studies on Mediterranean cyclones have mainly focused on winter, when most intense cyclones are expected to occur. However, intense Mediterranean cyclones have also taken place in summer and more studies are needed to address this issue. Dong et al. (2013), studied the summertime variability of the North Atlantic storm track and found that its dominant mode is closely associated with the negative phase of summer NAO. During this summer mode, a southward shift of the Atlantic storm track is linked with a weaker Mediterranean storm track and reduced precipitation over southern Europe, in contrast to its corresponding winter mode, in which the Mediterranean storm track is strengthened when the Atlantic jet is shifted south. Even though the summer NAO is weaker and confined to northern latitudes compared to its winter counterpart, it significantly affects precipitation in the Mediterranean which is anomalously wet during high summer NAO (Bladé et al. 2012) and relates to the relative cooling of the surface/lower troposphere, enhanced meridional circulation, and cloudiness over the eastern Mediterranean (Chronis et al. 2011)."*

*11. In section 4.4 it might be nice to note any relevant summary statements on Mediterranean cyclones from the recent IPCC AR6 report.*

The IPCC report is now referenced in the revised section 4.4:

*"This overall reduction of cyclones was also proposed by Raible et al. (2010) and Nissen et al. (2014) who used A2 and B2 SRES emission scenarios to show weaker rate of cyclones occurrences towards the end of the 21st century. Similar results were also confirmed by more recent studies that used simulation ensembles from CMIP5 (e.g. Zappa et al. 2015; Hochman et al. 2018). while the reduction of cyclones has been characterized as "robust" by the recent IPCC AR6 report (IPCC, 2021)."*

*12. 'Contrasting results' are noted in line 584, but is this too negative? From the following text there seems to be good agreement on an overall reduction in cyclones.*

Thank you for this comment. Indeed, contradictory results were seldom spotted in the conclusions of the cited papers. Section 4.4 has been substantially revised to synthesize the main results of previous studies. We mainly focus now on common contradictory results.

*13. The material on projected future trends should be accompanied by a brief review of climate model biases and fidelity in simulating the cyclones.*

We agree that this is an important aspect of recent and past climatological studies in the region. In the revised section 4.4, we have now included an introductory paragraph about the use of GCMs and RCMs in Mediterranean cyclone studies.

*14. Is the 'grand majority' of natural hazards on line 638 justified? What about droughts and wildfires, for example?*

We rephrased the sentence to :

*"A wide range of weather induced natural hazards in the Mediterranean is related to Mediterranean cyclones"*

*15. In the impacts section, is there any literature on impacts to shipping?*

Thank you for this comment. We could only find few studies that analyse the relationship between ship accidents and high sea-waves produced by cyclones. Nevertheless, a climatological study by Zhang and Li (2017) highlights the Mediterranean as an area of frequent ship accidents due to sea waves. Given the direct relationship between windstorms and cyclones in the region, we propose a more in-depth analysis on this issue. We added the following in section 5.2.2:

*"Few studies have analysed the impact of the sea state to ship accidents in the region. In particular, Bertotti and Cavaleri (2008) and Cavaleri et al. (2012) analyse two case studies where sea waves under the influence of intense Mediterranean cyclones were responsible for two ship accidents in February of 2005 and March 2010, respectively. In both cases, waves of more than 10 meters were reported to hit the vessels and cause substantial damages and human loss. More recently, Zhang and Li (2017) used the International Maritime Organization (IMO) database for ship accidents in the period 2001-2010 and showed that the Mediterranean Sea is one of the main regions in the globe with more frequent ship accidents induced by sea waves. Despite the direct relationship of cyclones to wind speed extremes in the region, a more thorough study is needed to better determine whether cyclones have indeed led to these accidents."*

Bertotti, L. and Cavaleri, L.: The predictability of the "Voyager" accident, Nat. Hazards Earth Syst. Sci., 8, 533–537, https://doi.org/10.5194/nhess-8-533-2008, 2008.

Cavaleri, L., Bertotti, L., Torrisi, L., Bitner-Gregersen, E., Serio, M., and Onorato, M.: Rogue waves in crossing seas: The Louis Majesty accident, J. Geophys. Res., 117, https://doi.org/10.1029/2012JC007923, 2012.

Zhang, Z. and Li, X.-M.: Global ship accidents and ocean swell-related sea states, Nat. Hazards Earth Syst. Sci., 17, 2041–2051, https://doi.org/10.5194/nhess-17-2041-2017, 2017.

*16. In Figure 7 the white and grey bars give quite different impressions on the seasonality - can this be explained?*

The grey bars correspond to high-impact weather events, produced by different atmospheric systems and subjectively selected during the MEDEX project. In the manuscript, the lower panel is only briefly commented and therefore we choose to remove it to avoid confusion to the reader.

**Annex**
The revised structure of the manuscript:

**1 Introduction**

**2 Climatological perspective of Mediterranean cyclones**
    2.1 Spatial variability and seasonal cycle of Mediterranean cyclone tracks
    2.2 Large-scale circulation and Mediterranean cyclones
    2.3 Current and future trends of Mediterranean cyclones

**3 Mediterranean cyclone dynamics**
    3.1 Large scale forcing of Mediterranean cyclogenesis
    3.2 Airstreams and fronts
    3.3 Role of diabatic processes in cyclones development
    3.4 Role of the orography in cyclone dynamics

**4 Subtypes of Mediterranean cyclones**
    4.1 Cyclones of similar physical characteristics
    4.2 The special case of medicanes: processes and classification
    4.3 The special case of medicanes: future trends

**5 Forecast challenges in Mediterranean cyclones**
    5.1 Dependence of predictability on processes taking place at different scales
    5.2 Predictability and forecasting strategies
    5.3 Use of observations and data assimilation
    5.4 New modelling opportunities and challenges

**6 Mediterranean cyclones as major environmental risks**
    6.1 Heavy precipitation and relevant impacts
    6.2 Wind-induced risks
        6.2.1 Windstorms
        6.2.2 Storm surges and sea waves
        6.2.3 Dust transport and episodes of particulate matter

**7 Open questions and research perspectives in the broader field of Mediterranean cyclones**
    7.1 Disentangling Mediterranean cyclone dynamics across spatial scales
    7.2 Identifying the specific characteristics of medicanes
    7.3 Understanding coupled processes in Mediterranean cyclones
    7.4 Solving resolution and parameterization issues for cyclone modelling
    7.5 Enhancing the use of observations and diagnostic tools
    7.6 Reducing error and uncertainty in numerical weather prediction

**8 Conclusion**